# Two-timescale Derivative Free Optimization for Performative Prediction with Markovian Data

## Abstract

This paper studies the performative prediction problem where a learner aims to minimize the expected loss with a decision-dependent data distribution. Such setting is motivated when outcomes can be affected by the prediction model, e.g., in strategic classification. We consider a state-dependent setting where the data distribution evolves according to an underlying controlled Markov chain. We focus on stochastic derivative free optimization (DFO) where the learner is given access to a loss function evaluation oracle with the above Markovian data. We propose a two-timescale DFO($\lambda$) algorithm that features (i) a sample accumulation mechanism that utilizes every observed sample to estimate the overall gradient of performative risk, and (ii) a two-timescale diminishing step size that balances the rates of DFO updates and bias reduction. Under a general non-convex optimization setting, we show that DFO($\lambda$) requires $\mathcal{O}(1/\epsilon^3)$ samples (up to a log factor) to attain a near-stationary solution with expected squared gradient norm less than $\epsilon > 0$. Numerical experiments verify our analysis.

## 1  Introduction

Consider the following stochastic optimization problem with decision-dependent data:

$$\min_{\boldsymbol{\theta} \in \mathbb{R}^d} \ \mathcal{L}(\boldsymbol{\theta}) = \mathbb{E}_{Z \sim \Pi_{\boldsymbol{\theta}}}\big[\ell(\boldsymbol{\theta}; Z)\big]. \tag{1}$$

Notice that the decision variable $\boldsymbol{\theta}$ appears in both the loss function $\ell(\boldsymbol{\theta}; Z)$ and the data distribution $\Pi_{\boldsymbol{\theta}}$ supported on Z. The overall loss function $\mathcal{L}(\boldsymbol{\theta})$ is known as the *performative risk* which captures the distributional shift due to changes in the deployed model. This setting is motivated by the recent studies on *performative prediction* (Perdomo et al., 2020), which considers outcomes that are supported by the deployed model $\boldsymbol{\theta}$ under training. For example, this models strategic classification (Hardt et al., 2016; Dong et al., 2018) in economical and financial practices such as with the training of loan classifier for customers who may react to the deployed model $\boldsymbol{\theta}$ to maximize their gains; or in price promotion mechanism (Zhang et al., 2018) where customers react to prices with the aim of gaining a lower price; or in ride sharing business (Narang et al., 2022) with customers who adjust their demand according to prices set by the platform.

The objective function $\mathcal{L}(\boldsymbol{\theta})$ is non-convex in general due to the effects of $\boldsymbol{\theta}$ on both the loss function and distribution. Numerous efforts have been focused on characterizing and finding the so-called *performative stable* solution which is a fixed point to the repeated risk minimization (RRM) process (Perdomo et al., 2020; Mendler-Dünner et al., 2020; Brown et al., 2022; Li & Wai, 2022; Roy et al., 2022; Drusvyatskiy & Xiao, 2022). While RRM might be a natural algorithm for scenarios when the learner is agnostic to the performative effects in the dynamic data distribution, the obtained solution maybe far from being optimal or stationary to (1).

Submitted to 37th Conference on Neural Information Processing Systems (NeurIPS 2023). Do not distribute.

On the other hand, recent works have studied *performative optimal* solutions that minimizes (1). This is challenging due to the non-convexity of $\mathcal{L}(\boldsymbol{\theta})$ and more importantly, the absence of knowledge of $\Pi_{\boldsymbol{\theta}}$. In fact, evaluating $\nabla\mathcal{L}(\boldsymbol{\theta})$ or its stochastic gradient estimate would require learning the distribution $\Pi_{\boldsymbol{\theta}}$ *a-priori* (Izzo et al., 2021). To design a tractable procedure, prior works have assumed structures for (1) such as approximating $\Pi_{\boldsymbol{\theta}}$ by Gaussian mixture (Izzo et al., 2021), $\Pi_{\boldsymbol{\theta}}$ depends linearly on $\boldsymbol{\theta}$ (Narang et al., 2022), etc., combined with a two-phase algorithm that separately learns $\Pi_{\boldsymbol{\theta}}$ and optimizes $\boldsymbol{\theta}$. Other works have assumed a *mixture dominance* structure (Miller et al., 2021) on the combined effect of $\Pi_{\boldsymbol{\theta}}$ and $\ell(\cdot)$ on $\mathcal{L}(\boldsymbol{\theta})$, which in turn implies that $\mathcal{L}(\boldsymbol{\theta})$ is convex. Based on this assumption, a derivative free optimization (DFO) algorithm was analyzed in Ray et al. (2022).

This paper focuses on approximating the *performative optimal* solution without relying on additional condition on the distribution $\Pi_{\boldsymbol{\theta}}$ and/or using a two-phase algorithm. We concentrate on stochastic DFO algorithms (Ghadimi & Lan, 2013) which do not involve first order information (i.e., gradient) about $\mathcal{L}(\boldsymbol{\theta})$. As an advantage, these algorithms avoid the need for estimating $\Pi_{\boldsymbol{\theta}}$. Instead, the learner is given access to the loss function evaluation oracle $\ell(\boldsymbol{\theta}; Z)$ and receive data samples from a controlled Markov chain. Note that the latter models the *stateful* and *strategic* agent setting considered in (Ray et al., 2022; Roy et al., 2022; Li & Wai, 2022; Brown et al., 2022).

| Stochastic DFO Settings | Rate |
| --- | --- |
| Decision-indep. (Ghadimi & Lan, 2013) | $\mathcal{O}(1/\epsilon^2)$ |
| Decision-depend. (Markov) | $\mathcal{O}(1/\epsilon^3)$ |

Table 1: Comparison of the expected convergence rates (to find an $\epsilon$-stationary point) for DFO under various settings where DFO is used to tackle an unstructured non-convex optimization problem such as (1).

Such setting is motivated when the actual data distribution adapts slowly to the decision model, which will be announced by the learner during the (stochastic) optimization process.

The proposed $\texttt{DFO}(\lambda)$ algorithm features (i) a two-timescale step sizes design to control the bias-variance tradeoff in the derivative-free gradient estimates, and (ii) a sample accumulation mechanism with forgetting factor $\lambda$ that aggregates every observed samples to control the amount of error in gradient estimates. In addition to the new algorithm design, our main findings are summarized below:

- Under the Markovian data setting, we show in Theorem 3.1 that the $\texttt{DFO}(\lambda)$ algorithm finds a near-stationary solution $\bar{\boldsymbol{\theta}}$ with $\mathbb{E}[\|\nabla\mathcal{L}(\bar{\boldsymbol{\theta}})\|^2] \leq \epsilon$ using $\mathcal{O}(\frac{d^2}{\epsilon^3}\log 1/\epsilon)$ samples/iterations. Compared to prior works, our analysis does not require structural assumption on the distribution $\Pi_{\boldsymbol{\theta}}$ or convexity condition on the performative risk (Izzo et al., 2021; Miller et al., 2021; Ray et al., 2022).
- Our analysis demonstrates the trade-off induced by the forgetting factor $\lambda$ in the $\texttt{DFO}(\lambda)$ algorithm. We identify the desiderata for the optimal value(s) of $\lambda$. We show that increasing $\lambda$ allows to reduce the number of samples requited by the algorithm if the performative risk gradient has a small Lipschitz constant.

For the rest of this paper, §2 describes the problem setup and the $\texttt{DFO}(\lambda)$ algorithm, §3 presents the main results, §4 outlines the proofs. Finally, we provide numerical results to verify our findings in §5.

Finally, as displayed in Table 1, we remark that stochastic DFO under *decision dependent* (and Markovian) samples has a convergence rate of $\mathcal{O}(1/\epsilon^3)$ towards an $\epsilon$-stationary point, which is worse than the decision independent setting that has $\mathcal{O}(1/\epsilon^2)$ in Ghadimi & Lan (2013). We believe that this is a fundamental limit for DFO-type algorithms when tackling problems with decision-dependent sample due to the challenges in designing a low variance gradient estimator; see §4.1.

**Related Works**. The idea of DFO dates back to Nemirovskiĭ (1983), and has been extensively studied thereafter Flaxman et al. (2005); Agarwal et al. (2010); Nesterov & Spokoiny (2017); Ghadimi & Lan (2013). Results on matching lower bound were established in (Jamieson et al., 2012). While a similar DFO framework is adopted in the current paper for performative prediction, our algorithm is limited to using a special design in the gradient estimator to avoid introducing unwanted biases.

There are only a few works considering the Markovian data setting in performative prediction. Brown et al. (2022) is the first paper to study the dynamic settings, where the response of agents to learner's deployed classifier is modeled as a function of classifier and the current distribution of the population; also see (Izzo et al., 2022). On the other hand, Li & Wai (2022); Roy et al. (2022) model the unforgetful nature and the reliance on past experiences of *single/batch* agent(s) via controlled Markov Chain. Lastly, Ray et al. (2022) investigated the state-dependent framework where agents' response may be driven to best response at a geometric rate.

---

**Algorithm 1** DFO $(\lambda)$ Algorithm

---

1: **Input:** Constants $\delta_0, \eta_0, \tau_0, \alpha, \beta$, maximum epochs $T$, forgetting factor $\lambda$, loss function $\ell(\cdot; \cdot)$.

2: **Initialization:** Set initial $\boldsymbol{\theta}_0$ and sample $Z_0$.

3: **for** $k = 0$ **to** $T - 1$ **do**

4:     $\delta_k \leftarrow \delta_0/(1+k)^\beta$, $\eta_k \leftarrow \eta_0/(1+k)^\alpha$, $\tau_k \leftarrow \max\{1, \tau_0 \log(1+k)\}$

5:     Update $\boldsymbol{\theta}_k^{(1)} \leftarrow \boldsymbol{\theta}_k$, $Z_k^{(0)} \leftarrow Z_k$, $\boldsymbol{u}_k \sim \text{Unif}(\mathbb{S}^{d-1})$

6:     **for** $m = 1, 2, \cdots, \tau_k$ **do**

7:         Deploy the model $\check{\boldsymbol{\theta}}_k^{(m)} = \boldsymbol{\theta}_k^{(m)} + \delta_k \boldsymbol{u}_k$

8:         Draw $Z_k^{(m)} \sim \mathbb{T}_{\check{\boldsymbol{\theta}}_k^{(m)}}(Z_k^{(m-1)}, \cdot)$

9:         Update $\boldsymbol{\theta}_k^{(m)}$ as

$$\boldsymbol{g}_k^{(m)} = \frac{d}{\delta_k} \ell(\check{\boldsymbol{\theta}}_k^{(m)}; Z_k^{(m)}) \boldsymbol{u}_k,$$

$$\boldsymbol{\theta}_k^{(m+1)} = \boldsymbol{\theta}_k^{(m)} - \eta_k \lambda^{\tau_k - m} \boldsymbol{g}_k^{(m)}.$$

10:     **end for**

11:     $Z_{k+1} \leftarrow Z_k^{(\tau_k)}$, $\boldsymbol{\theta}_{k+1} \leftarrow \boldsymbol{\theta}_k^{(\tau_k+1)}$.

12: **end for**

   **Output:** Last iterate $\boldsymbol{\theta}_T$.

---

**Notations**: Let $\mathbb{R}^d$ be the $d$-dimensional Euclidean space equipped with inner product $\langle \cdot, \cdot \rangle$ and induced norm $\|x\| = \sqrt{\langle x, x \rangle}$. Let $\mathcal{S}$ be a (measurable) sample space, and $\mu, \nu$ are two probability measures defined on $\mathcal{S}$. Then, we use $\boldsymbol{\delta}_{\text{TV}}(\mu, \nu) := \sup_{A \subset \mathcal{S}} \mu(A) - \nu(A)$ to denote the total variation distance between $\mu$ and $\nu$. Denote $\mathbb{T}_{\boldsymbol{\theta}}(\cdot, \cdot)$ as the state-dependent Markov kernel and its stationary distribution is $\Pi_{\boldsymbol{\theta}}(\cdot)$. Let $\mathbb{B}^d$ and $\mathbb{S}^{d-1}$ be the unit ball and its boundary (i.e., a unit sphere) centered around the origin in $d$-dimensional Euclidean space, respectively, and correspondingly, the ball and sphere of radius $r > 0$ are $r\mathbb{B}^d$ and $r\mathbb{S}^{d-1}$.

## 2 Problem Setup and Algorithm Design

In this section, we develop the DFO $(\lambda)$ algorithm for tackling (1) and describe the problem setup. Assume that $\mathcal{L}(\boldsymbol{\theta})$ is differentiable, we focus on finding an $\epsilon$-*stationary* solution, $\boldsymbol{\theta}$, which satisfies

$$\|\nabla \mathcal{L}(\boldsymbol{\theta})\|^2 \leq \epsilon. \tag{2}$$

With the goal of reaching (2), there are two key challenges in our stochastic algorithm design: (i) to estimate the gradient $\nabla \mathcal{L}(\boldsymbol{\theta})$, and (ii) to handle the *stateful* setting where one cannot draw samples directly from the distribution $\Pi_{\boldsymbol{\theta}}$. We shall discuss how the proposed DFO $(\lambda)$ algorithm, which is summarized in Algorithm 1, tackles the above issues through utilizing two ingredients: (a) two-timescales step sizes, and (b) sample accumulation with the forgetting factor $\lambda \in [0, 1)$.

**Estimating $\nabla \mathcal{L}(\boldsymbol{\theta})$ via Two-timescales DFO.** First notice that the gradient of $\mathcal{L}(\cdot)$ can be derived as

$$\nabla \mathcal{L}(\boldsymbol{\theta}) = \mathbb{E}_{Z \sim \Pi_{\boldsymbol{\theta}}}[\nabla \ell(\boldsymbol{\theta}; Z) + \ell(\boldsymbol{\theta}; Z) \nabla_{\boldsymbol{\theta}} \log \Pi_{\boldsymbol{\theta}}(Z)], \tag{3}$$

As a result, constructing the stochastic estimates of $\nabla \mathcal{L}(\boldsymbol{\theta})$ typically requires knowledge of $\Pi_{\boldsymbol{\theta}}(\cdot)$ which may not be known a-priori unless a separate estimation procedure is applied; see e.g., (Izzo et al., 2021). To avoid the need for direct evaluations of $\nabla_{\boldsymbol{\theta}} \log \Pi_{\boldsymbol{\theta}}(Z)$, we consider an alternative design via zero-th order optimization (Ghadimi & Lan, 2013). The intuition comes from observing that with $\delta \to 0^+$, $\mathcal{L}(\boldsymbol{\theta} + \delta \boldsymbol{u}) - \mathcal{L}(\boldsymbol{\theta})$ is an approximate of the directional derivative of $\mathcal{L}$ along $\boldsymbol{u}$. This suggests that an estimate for $\nabla \mathcal{L}(\boldsymbol{\theta})$ can be constructed using the *objective function values* of $\ell(\boldsymbol{\theta}; Z)$ only.

Inspired by the above, we aim to construct a gradient estimate by querying $\ell(\cdot)$ at randomly perturbed points. Formally, given the current iterate $\boldsymbol{\theta} \in \mathbb{R}^d$ and a query radius $\delta > 0$, we sample a vector $\boldsymbol{u} \in \mathbb{R}^d$ uniformly from $\mathbb{S}^{d-1}$. The zero-th order gradient estimator for $\mathcal{L}(\boldsymbol{\theta})$ is then defined as

$$g_\delta(\boldsymbol{\theta}; \boldsymbol{u}, Z) := \frac{d}{\delta} \ell(\check{\boldsymbol{\theta}}; Z) \boldsymbol{u} \quad \text{with} \quad \check{\boldsymbol{\theta}} := \boldsymbol{\theta} + \delta \boldsymbol{u}, \ Z \sim \Pi_{\check{\boldsymbol{\theta}}}(\cdot). \tag{4}$$

In fact, as $\boldsymbol{u}$ is zero-mean, $g_\delta(\boldsymbol{\theta}; \boldsymbol{u}, Z)$ is an unbiased estimator for $\nabla \mathcal{L}_\delta(\boldsymbol{\theta})$. Here, $\mathcal{L}_\delta(\boldsymbol{\theta})$ is a smooth approximation of $\mathcal{L}(\boldsymbol{\theta})$ (Flaxman et al., 2005; Nesterov & Spokoiny, 2017) defined as

$$\mathcal{L}_\delta(\boldsymbol{\theta}) = \mathbb{E}_{\boldsymbol{u}}[\mathcal{L}(\check{\boldsymbol{\theta}})] = \mathbb{E}_{\boldsymbol{u}}[\mathbb{E}_{Z \sim \Pi_{\check{\boldsymbol{\theta}}}}[\ell(\check{\boldsymbol{\theta}}; Z)]]. \tag{5}$$

Furthermore, it is known that under mild condition [cf. Assumption 3.1 to be discussed later], $\|\nabla \mathcal{L}_\delta(\boldsymbol{\theta}) - \nabla \mathcal{L}(\boldsymbol{\theta})\| = \mathcal{O}(\delta)$ and thus (4) is an $\mathcal{O}(\delta)$-biased estimate for $\nabla \mathcal{L}(\boldsymbol{\theta})$.

We remark that the gradient estimator in (4) differs from the one used in classical works on DFO such as (Ghadimi & Lan, 2013). The latter takes the form of $\frac{d}{\delta}(\ell(\check{\boldsymbol{\theta}}; Z) - \ell(\boldsymbol{\theta}; Z))\boldsymbol{u}$. Under the setting of standard stochastic optimization where the sample $Z$ is drawn *independently* of $\boldsymbol{u}$ and Lipschitz continuous $\ell(\cdot; Z)$, the said estimator in (Ghadimi & Lan, 2013) is shown to have constant variance while it remains $\mathcal{O}(\delta)$-biased. Such properties *cannot* be transferred to (4) since $Z$ is drawn from a distribution dependent on $\boldsymbol{u}$ via $\check{\boldsymbol{\theta}} = \boldsymbol{\theta} + \delta\boldsymbol{u}$. In this case, the two-point gradient estimator would become biased; see §4.1.

However, we note that the variance of (4) would increase as $\mathcal{O}(1/\delta^2)$ when $\delta \to 0$, thus the parameter $\delta$ yields a bias-variance trade off in the estimator design. To remedy for the increase of variance, the DFO$(\lambda)$ algorithm incorporates a *two-timescale step size* design for generating gradient estimates $(\delta_k)$ and updating models $(\eta_k)$, respectively. Our design principle is such that the models are updated at a *slower timescale* to adapt to the gradient estimator with $\mathcal{O}(1/\delta^2)$ variance. Particularly, we will set $\eta_{k+1}/\delta_{k+1} \to 0$ to handle the bias-variance trade off, e.g., by setting $\alpha > \beta$ in line 4 of Algorithm 1.

**Markovian Data and Sample Accumulation.** We consider a setting where the sample/data distribution observed by the DFO$(\lambda)$ algorithm evolves according to a *controlled Markov chain (MC)*. Notice that this describes a stateful agent(s) scenario such that the deployed models $(\boldsymbol{\theta})$ would require time to manifest their influence on the samples obtained; see (Li & Wai, 2022; Roy et al., 2022; Brown et al., 2022; Ray et al., 2022; Izzo et al., 2022).

To describe the setting formally, we denote $\mathbb{T}_{\boldsymbol{\theta}} : \mathsf{Z} \times \mathcal{Z} \to \mathbb{R}_+$ as a Markov kernel controlled by a deployed model $\boldsymbol{\theta}$. For a given $\boldsymbol{\theta}$, the kernel has a unique stationary distribution $\Pi_{\boldsymbol{\theta}}(\cdot)$. Under this setting, suppose that the previous state/sample is $Z$, the next sample follows the distribution $Z' \sim \mathbb{T}_{\boldsymbol{\theta}}(Z, \cdot)$ which is not necessarily the same as $\Pi_{\boldsymbol{\theta}}(\cdot)$. As a consequence, the gradient estimator (4) is not an unbiased estimator of $\nabla \mathcal{L}_{\delta}(\boldsymbol{\theta})$ since $Z \sim \Pi_{\check{\boldsymbol{\theta}}}(\cdot)$ cannot be conveniently accessed.

A common strategy in settling the above issue is to allow a *burn-in* phase in the algorithm as in (Ray et al., 2022); also commonly found in MCMC methods (Robert et al., 1999). Using the fact that $\mathbb{T}_{\boldsymbol{\theta}}$ admits the stationary distribution $\Pi_{\boldsymbol{\theta}}$, if one can wait a sufficiently long time before applying the current sample, i.e., consider initializing with the previous sample $Z^{(0)} = Z$, the procedure

$$Z^{(m)} \sim \mathbb{T}_{\boldsymbol{\theta}}(Z^{(m-1)}, \cdot), \ m = 1, \ldots, \tau, \tag{6}$$

would yield a sample $Z^+ = Z^{(\tau)}$ that admits a distribution close to $\Pi_{\boldsymbol{\theta}}$ provided that $\tau \gg 1$ is sufficiently large compared to the mixing time of $\mathbb{T}_{\boldsymbol{\theta}}$.

Intuitively, the procedure (6) may be inefficient as a number of samples $Z^{(1)}, Z^{(2)}, \ldots, Z^{(\tau-1)}$ will be completely ignored at the end of each iteration. As a remedy, the DFO$(\lambda)$ algorithm incorporates a sample accumulation mechanism which gathers the gradient estimates generated from possibly non-stationary samples via a forgetting factor of $\lambda \in [0, 1)$. Following (4), $\nabla \mathcal{L}(\boldsymbol{\theta})$ is estimated by

$$\boldsymbol{g} = \frac{d}{\delta} \sum_{m=1}^{\tau} \lambda^{\tau-m} \ell(\boldsymbol{\theta}^{(m)} + \delta\boldsymbol{u}; Z^{(m)}) \boldsymbol{u}, \ \text{with} \ Z^{(m)} \sim \mathbb{T}_{\boldsymbol{\theta}^{(m)} + \delta\boldsymbol{u}}(Z^{(m-1)}, \cdot). \tag{7}$$

At a high level, the mechanism works by assigning large weights to samples that are close to the end of an epoch (which are less biased). Moreover, $\boldsymbol{\theta}^{(m)}$ is *simultaneously updated* within the epoch to obtain an online algorithm that gradually improves the objective value of (1). Note that with $\lambda = 0$, the DFO(0) algorithm reduces into one that utilizes *burn-in* (6). We remark that from the implementation perspective for performative prediction, Algorithm 1 corresponds to a *greedy deployment* scheme (Perdomo et al., 2020) as the latest model $\boldsymbol{\theta}_k^{(m)} + \delta_k \boldsymbol{u}_k$ is deployed at every sampling step. Line 6–10 of Algorithm 1 details the above procedure.

Lastly, we note that recent works have analyzed stochastic algorithms that rely on a *single trajectory* of samples taken from a Markov Chain, e.g., (Sun et al., 2018; Karimi et al., 2019; Doan, 2022), that are based on stochastic gradient. Sun & Li (2019) considered a DFO algorithm for general optimization problems but the MC studied is not controlled by $\boldsymbol{\theta}$.

# 3 Main Results

This section studies the convergence of the DFO$(\lambda)$ algorithm and demonstrates that the latter finds an $\epsilon$-stationary solution [cf. (2)] to (1). We first state the assumptions required for our analysis:

**Assumption 3.1. (Smoothness)** $\mathcal{L}(\boldsymbol{\theta})$ is differentiable, and there exists a constant $L > 0$ such that

$$\|\nabla \mathcal{L}(\boldsymbol{\theta}) - \nabla \mathcal{L}(\boldsymbol{\theta}')\| \leq L \|\boldsymbol{\theta} - \boldsymbol{\theta}'\|, \ \forall \boldsymbol{\theta}, \boldsymbol{\theta}' \in \mathbb{R}^d.$$

**Assumption 3.2. (Bounded Loss)** There exists a constant $G > 0$ such that

$$|\ell(\boldsymbol{\theta}; z)| \leq G, \ \forall \, \boldsymbol{\theta} \in \mathbb{R}^d, \ \forall \, z \in \mathsf{Z}.$$

**Assumption 3.3. (Lipschitz Distribution Map)** There exists a constant $L_1 > 0$ such that

$$\boldsymbol{\delta}_{\mathrm{TV}} \left( \Pi_{\boldsymbol{\theta}_1}, \Pi_{\boldsymbol{\theta}_2} \right) \leq L_1 \left\| \boldsymbol{\theta}_1 - \boldsymbol{\theta}_2 \right\| \quad \forall \boldsymbol{\theta}_1, \boldsymbol{\theta}_2 \in \mathbb{R}^d.$$

The conditions above state that the gradient of the performative risk is Lipschitz continuous and the state-dependent distribution vary smoothly w.r.t. $\boldsymbol{\theta}$. Note that Assumption 3.1 is found in recent works such as (Izzo et al., 2021; Ray et al., 2022), and Assumption 3.2 can be found in (Izzo et al., 2021). Assumption 3.3 is slightly strengthened from the Wasserstein-1 distance bound in (Perdomo et al., 2020), and it gives better control for distribution shift in our Markovian data setting.

Next, we consider the assumptions about the controlled Markov chain induced by $\mathbb{T}_{\boldsymbol{\theta}}$:

**Assumption 3.4. (Geometric Mixing)** Let $\{Z_k\}_{k \geq 0}$ denote a Markov Chain on the state space $\mathsf{Z}$ with transition kernel $\mathbb{T}_{\boldsymbol{\theta}}$ and stationary measure $\Pi_{\boldsymbol{\theta}}$. There exist constants $\rho \in [0, 1)$, $M \geq 0$, such that for any $k \geq 0$, $z \in \mathsf{Z}$,

$$\boldsymbol{\delta}_{\mathrm{TV}} \left( \mathbb{P}_{\boldsymbol{\theta}}(Z_k \in \cdot | Z_0 = z), \Pi_{\boldsymbol{\theta}} \right) \leq M \rho^k.$$

**Assumption 3.5. (Smoothness of Markov Kernel)** There exists a constant $L_2 \geq 0$ such that

$$\boldsymbol{\delta}_{\mathrm{TV}} \left( \mathbb{T}_{\boldsymbol{\theta}_1}(z, \cdot), \mathbb{T}_{\boldsymbol{\theta}_2}(z, \cdot) \right) \leq L_2 \left\| \boldsymbol{\theta}_1 - \boldsymbol{\theta}_2 \right\|, \ \forall \boldsymbol{\theta}_1, \boldsymbol{\theta}_2 \in \mathbb{R}^d, \ z \in \mathsf{Z}.$$

Assumption 3.4 is a standard condition on the mixing time of the Markov chain induced by $\mathbb{T}_{\boldsymbol{\theta}}$; Assumption 3.5 imposes a smoothness condition on the Markov transition kernel $\mathbb{T}_{\boldsymbol{\theta}}$ with respect to $\boldsymbol{\theta}$. For instance, the geometric dynamically environment in Ray et al. (2022) constitutes a special case which satisfies the above conditions.

Unlike (Ray et al., 2022; Izzo et al., 2021; Miller et al., 2021), we do not impose any additional assumption (such as mixture dominance) other than Assumption 3.3 on $\Pi_{\boldsymbol{\theta}}$. As a result, (1) remains an 'unstructured' non-convex optimization problem. Our main theoretical result on the convergence of the DFO $(\lambda)$ algorithm towards a near-stationary solution of (1) is summarized as:

---

**Theorem 3.1.** *Suppose Assumptions 3.1-3.5 hold, step size sequence $\{\eta_k\}_{k \geq 1}$, and query radius sequence $\{\delta_k\}_{k \geq 1}$ satisfy the following conditions,*

$$\eta_k = d^{-2/3} \cdot (1 + k)^{-2/3}, \quad \delta_k = d^{1/3} \cdot (1 + k)^{-1/6},$$

$$\tau_k = \max\{1, \frac{2}{\log 1 / \max\{\rho, \lambda\}} \log(1 + k)\} \quad \forall k \geq 0. \tag{8}$$

*Then, there exists constants $t_0, c_5, c_6, c_7$, such that for any $T \geq t_0$, the iterates $\{\boldsymbol{\theta}_k\}_{k \geq 0}$ generated by DFO $(\lambda)$ satisfy the following inequality,*

$$\min_{0 \leq k \leq T} \mathbb{E} \left\| \nabla \mathcal{L}(\boldsymbol{\theta}_k) \right\|^2 \leq 12 \max \left\{ c_5(1 - \lambda), c_6, \frac{c_7}{1 - \lambda} \right\} \frac{d^{2/3}}{(T + 1)^{1/3}}. \tag{9}$$

---

We have defined the following quantities and constants:

$$c_5 = 2G, \quad c_6 = \frac{\max\{L^2, G^2(1 - \beta)\}}{1 - 2\beta}, \quad c_7 = \frac{LG^2}{2\beta - \alpha + 1}, \tag{10}$$

with $\alpha = \frac{2}{3}, \beta = \frac{1}{6}$. Observe the following corollary on the iteration complexity of DFO $(\lambda)$ algorithm:

**Corollary 3.1.** *($\epsilon$-stationarity) Suppose that the Assumptions of Theorem 3.1 hold. Fix any $\epsilon > 0$, the condition $\min_{0 \leq k \leq T-1} \mathbb{E} \left\| \nabla \mathcal{L}(\boldsymbol{\theta}_k) \right\|^2 \leq \epsilon$ holds whenever*

$$T \geq \left( 12 \max \left\{ c_5(1 - \lambda), c_6, \frac{c_7}{1 - \lambda} \right\} \right)^3 \frac{d^2}{\epsilon^3}. \tag{11}$$

In the corollary above, the lower bound on $T$ is expressed in terms of the number of epochs that Algorithm 1 needs to achieve the target accuracy. Consequently, the total number of samples required (i.e., the number of inner iterations taken in Line 6–9 of Algorithm 1 across all epochs) is:

$$\mathsf{S}_{\epsilon} = \sum_{k=1}^{T} \tau_k = \mathcal{O} \left( \frac{d^2}{\epsilon^3} \log(1/\epsilon) \right). \tag{12}$$

We remark that due to the decision-dependent properties of the samples, the $\mathtt{DFO}\,(\lambda)$ algorithm exhibits a worse sampling complexity (12) than prior works in stochastic DFO algorithm, e.g., (Ghadimi & Lan, 2013) which shows a rate of $\mathcal{O}(d/\epsilon^2)$ on non-convex smooth objective functions. In particular, the adopted one-point gradient estimator in (4) admits a variance that can only be controlled by a time varying $\delta$; see the discussions in §4.1.

Achieving the desired convergence rate requires setting $\eta_k = \Theta(k^{-2/3})$, $\delta_k = \Theta(k^{-1/6})$, i.e., yielding a two-timescale step sizes design with $\eta_k/\delta_k \to 0$. Notice that the influence of forgetting factor $\lambda$ are reflected in the constant factor of (9). Particularly, if $c_5 > c_7$ and $c_5 \geq c_6$, the optimal choice is $\lambda = 1 - \sqrt{\frac{c_7}{c_5}}$, otherwise the optimal choice is $\lambda \in [0, 1 - c_7/c_6]$. Informally, this indicates that when the performative risk is smoother (i.e. its gradient has a small Lipschitz constant), a large $\lambda$ can speed up the convergence of the algorithm; otherwise a smaller $\lambda$ is preferable.

## 4 Proof Outline of Main Results

This section outlines the key steps in proving Theorem 3.1. Notice that analyzing the $\mathtt{DFO}\,(\lambda)$ algorithm is challenging due to the two-timescales step sizes and Markov chain samples with time varying kernel. Our analysis departs significantly from prior works such as (Ray et al., 2022; Izzo et al., 2021; Brown et al., 2022; Li & Wai, 2022) to handle the challenges above.

Let $\mathcal{F}^k = \sigma(\boldsymbol{\theta}_0, Z_s^{(m)}, u_s, 0 \leq s \leq k, 0 \leq m \leq \tau_k)$ be the filtration. Our first step is to exploit the smoothness of $\mathcal{L}(\boldsymbol{\theta})$ to bound the squared norms of gradient. Observe that:

**Lemma 4.1. (Decomposition)** *Under Assumption 3.1, it holds that*

$$\sum_{k=0}^{t} \mathbb{E}\left\| \nabla \mathcal{L}(\boldsymbol{\theta}_k) \right\|^2 \leq \mathbf{I}_1(t) + \mathbf{I}_2(t) + \mathbf{I}_3(t) + \mathbf{I}_4(t), \tag{13}$$

*for any $t \geq 1$, where*

$$\mathbf{I}_1(t) := \sum_{k=1}^{t} \frac{1-\lambda}{\eta_k} \left( \mathbb{E}\left[\mathcal{L}(\boldsymbol{\theta}_k)\right] - \mathbb{E}\left[\mathcal{L}(\boldsymbol{\theta}_{k+1})\right] \right)$$

$$\mathbf{I}_2(t) := -\sum_{k=1}^{t} \mathbb{E} \left\langle \nabla \mathcal{L}(\boldsymbol{\theta}_k) \Big| (1-\lambda) \sum_{m=1}^{\tau_k} \lambda^{\tau_k - m} \cdot \left( g_k^{(m)} - \mathbb{E}_{Z \sim \Pi_{\check{\boldsymbol{\theta}}_k}}\left[ g_{\delta_k}(\boldsymbol{\theta}_k; u_k, Z) \right] \right) \right\rangle$$

$$\mathbf{I}_3(t) := -\sum_{k=1}^{t} \mathbb{E} \left\langle \nabla \mathcal{L}(\boldsymbol{\theta}_k) \Big| (1-\lambda) \left( \sum_{m=1}^{\tau_k} \lambda^{\tau_k - m} \nabla \mathcal{L}_{\delta_k}(\boldsymbol{\theta}_k) \right) - \nabla \mathcal{L}(\boldsymbol{\theta}_k) \right\rangle$$

$$\mathbf{I}_4(t) := \frac{L(1-\lambda)}{2} \sum_{k=1}^{t} \eta_k \mathbb{E} \left\| \sum_{m=1}^{\tau_k} \lambda^{\tau_k - m} g_k^{(m)} \right\|^2$$

The lemma is achieved through the standard descent lemma implied by Assumption 3.1 and decomposing the upper bound on $\|\nabla \mathcal{L}(\boldsymbol{\theta}_k)\|^2$ into respectful terms; see the proof in Appendix A. Among the terms on the right hand side of (13), we note that $\mathbf{I}_1(t), \mathbf{I}_3(t)$ and $\mathbf{I}_4(t)$ arises directly from Assumption 3.1, while $\mathbf{I}_2(t)$ comes from bounding the noise terms due to Markovian data.

We bound the four components in Lemma 4.1 as follows. For simplicity, we denote $\mathcal{A}(t) := \frac{1}{1+t} \sum_{k=0}^{t} \mathbb{E}\left\| \nabla \mathcal{L}(\boldsymbol{\theta}_k) \right\|^2$. Among the four terms, we highlight that the main challenge lies on obtaining a tight bound for $\mathbf{I}_2(t)$. Observe that

$$\mathbf{I}_2(t) \leq (1-\lambda)\mathbb{E}\left[ \sum_{k=0}^{t} \left\| \nabla \mathcal{L}(\boldsymbol{\theta}_k) \right\| \cdot \left\| \sum_{m=1}^{\tau_k} \lambda^{\tau_k - m} \Delta_{k,m} \right\| \right] \tag{14}$$

where $\Delta_{k,m} \stackrel{\text{def}}{=} \mathbb{E}_{\mathcal{F}^{k-1}}[g_k^{(m)} - \mathbb{E}_{Z \sim \Pi_{\check{\boldsymbol{\theta}}_k}} g_k(\boldsymbol{\theta}_k; u_k, Z)]$. There are two sources of bias in $\Delta_{k,m}$: one is the noise induced by drifting of decision variable in every epoch, the other is the bias that depends on the mixing time of Markov kernel. To control these biases, we are inspired by the proof of (Wu et al., 2020, Theorem 4.7) to introduce a reference Markov chain $\tilde{Z}_k^{(\ell)}$, $\ell = 0, ..., \tau_k$, whose decision variables remains fixed for a period of length $\tau_k$ and is initialized with $\tilde{Z}_k^{(0)} = Z_k^{(0)}$:

$$\tilde{Z}_k^{(0)} \xrightarrow{\check{\boldsymbol{\theta}}_k} \tilde{Z}_k^{(1)} \xrightarrow{\check{\boldsymbol{\theta}}_k} \tilde{Z}_k^{(2)} \xrightarrow{\check{\boldsymbol{\theta}}_k} \tilde{Z}_k^{(3)} \cdots \xrightarrow{\check{\boldsymbol{\theta}}_k} \tilde{Z}_k^{(\tau_k)} \tag{15}$$

and we recall that the actual chain in the algorithm evolves as

$$Z_k^{(0)} \xrightarrow{\check{\boldsymbol{\theta}}_{k+1}^{(0)}} Z_k^{(1)} \xrightarrow{\check{\boldsymbol{\theta}}_{k+1}^{(1)}} Z_k^{(2)} \cdots \xrightarrow{\check{\boldsymbol{\theta}}_{k+1}^{(\tau_k - 1)}} Z_k^{(\tau_k)}. \tag{16}$$

With the help of the reference chain, we decompose $\Delta_{k,m}$ into

$$\Delta_{k,m} = \mathbb{E}_{\mathcal{F}^{k-1}}\left[\frac{d}{\delta_k}\left(\mathbb{E}[\ell(\check{\boldsymbol{\theta}}_k^{(m)}; Z_k^{(m)})|\check{\boldsymbol{\theta}}_k^{(m)}, Z_k^{(0)}] - \mathbb{E}_{\tilde{Z}_k^{(m)}}[\ell(\check{\boldsymbol{\theta}}_k^{(m)}; \tilde{Z}_k^{(m)})|\check{\boldsymbol{\theta}}_k^{(m)}, \tilde{Z}_k^{(0)}]\right)u_k\right]$$

$$+ \mathbb{E}_{\mathcal{F}^{k-1}}\left[\frac{d}{\delta_k}\left(\mathbb{E}_{\tilde{Z}_k^{(m)}}[\ell(\check{\boldsymbol{\theta}}_k^{(m)}; \tilde{Z}_k^{(m)})|\check{\boldsymbol{\theta}}_k^{(m)}, \tilde{Z}_k^{(0)}] - \mathbb{E}_{Z\sim\Pi_{\check{\boldsymbol{\theta}}_k}}[\ell(\check{\boldsymbol{\theta}}_k^{(m)}; Z)|\check{\boldsymbol{\theta}}_k^{(m)}]\right)u_k\right]$$

$$+ \mathbb{E}_{\mathcal{F}^{k-1}}\frac{d}{\delta_k}\mathbb{E}_{Z\sim\Pi_{\check{\boldsymbol{\theta}}_k}}\left[\ell(\check{\boldsymbol{\theta}}_k^{(m)}; Z) - \ell(\check{\boldsymbol{\theta}}_k; Z)|\check{\boldsymbol{\theta}}_k^{(m)}, \check{\boldsymbol{\theta}}_k\right]u_k := A_1 + A_2 + A_3$$

We remark that $A_1$ reflects the drift of (16) from initial sample $Z_k^{(0)}$ driven by varying $\check{\boldsymbol{\theta}}_k^{(m)}$, $A_2$ captures the statistical discrepancy between above two Markov chains (16) and (15) at same step $m$, and $A_3$ captures the drifting gap between $\check{\boldsymbol{\theta}}_k$ and $\check{\boldsymbol{\theta}}_k^{(m)}$. Applying Assumption 3.3, $A_1$ and $A_2$ can be upper bounded with the smoothness and geometric mixing property of Markov kernel. In addition, $A_3$ can be upper bounded using Lipschitz condition on (stationary) distribution map $\Pi_{\boldsymbol{\theta}}$. Finally, the forgetting factor $\lambda$ helps to control $\|\check{\boldsymbol{\theta}}_k^{(\cdot)} - \check{\boldsymbol{\theta}}_k\|$ to be at the same order of a single update. Therefore, $\|\Delta_{k,m}\|$ can be controlled by an upper bound relying on $\lambda, \rho, L$.

The following lemma summarizes the above results as well as the bounds on the other terms:

**Lemma 4.2.** *Under Assumption 3.2, 3.3, 3.4 and 3.5, with $\eta_{t+1} = \eta_0(1+t)^{-\alpha}$, $\delta_{t+1} = \delta_0(1+t)^{-\beta}$ and $\alpha \in (0,1)$, $\beta \in (0, \frac{1}{2})$. Suppose that $0 < 2\alpha - 4\beta < 1$ and*

$$\tau_k \geq \frac{1}{\log 1/\max\{\rho,\lambda\}}\left(\log(1+k) + \max\{\log\frac{\delta_0}{d}, 0\}\right).$$

*Then, it holds that*

$$\mathbf{I}_2(t) \leq \frac{c_2 d^{5/2}}{(1-\lambda)^2}\mathcal{A}(t)^{\frac{1}{2}}(1+t)^{1-(\alpha-2\beta)}, \quad \forall t \geq \max\{t_1, t_2\} \tag{17}$$

$$\mathbf{I}_1(t) \leq c_1(1-\lambda)(1+t)^{\alpha}, \quad \mathbf{I}_3(t) \leq c_3\mathcal{A}(t)^{\frac{1}{2}}(1+t)^{1-\beta}, \quad \mathbf{I}_4(t) \leq \frac{c_4 d^2}{1-\lambda}(1+t)^{1-(\alpha-2\beta)}, \tag{18}$$

*where $t_1, t_2$ are defined in (25), (26), and $c_1, c_2, c_3, c_4$ are constants defined as follows:*

$$c_1 := 2G/\eta_0, \quad c_2 := \frac{\eta_0}{\delta_0^2}\frac{6\cdot(L_1 G^2 + L_2 G^2 + \sqrt{L}G^{3/2})}{\sqrt{1-2\alpha+4\beta}},$$

$$c_3 := \frac{2}{\sqrt{1-2\beta}}\max\{L\delta_0, G\sqrt{1-\beta}\}, \quad c_4 := \frac{\eta_0}{\delta_0^2}\cdot\frac{LG^2}{2\beta-\alpha+1}.$$

See Appendix B for the proof. We comment that the bound for $\mathbf{I}_4(t)$ cannot be improved. As a concrete example, consider the constant function $\ell(\boldsymbol{\theta}; z) = c \neq 0$ for all $z \in \mathsf{Z}$, it can be shown that $\|g_k^{(m)}\|^2 = c^2$ and consequently $\mathbf{I}_4(t) = \Omega(\eta_k/\delta_k^2) = \Omega(t^{1-(\alpha-2\beta)})$, which matches (18). Finally, plugging Lemma 4.2 into Lemma 4.1 gives:

$$\mathcal{A}(t) \leq \frac{c_1(1-\lambda)}{(1+t)^{1-\alpha}} + \frac{c_2 d^{5/2}}{(1-\lambda)^2}\frac{\mathcal{A}(t)^{\frac{1}{2}}}{(1+t)^{\alpha-2\beta}} + c_3\frac{\mathcal{A}(t)^{\frac{1}{2}}}{(1+t)^{\beta}} + c_4\frac{d^2}{1-\lambda}\frac{1}{(1+t)^{\alpha-2\beta}}. \tag{19}$$

Since $\mathcal{A}(t) \geq 0$, the above is a quadratic inequality that implies the following bound:

**Lemma 4.3.** *Under Assumption 3.1–3.5, with the step sizes $\eta_{t+1} = \eta_0(1+t)^{-\alpha}$, $\delta_{t+1} = \delta_0(1+t)^{-\beta}$, $\tau_k \geq \frac{1}{\log 1/\max\{\rho,\lambda\}}\left(\log(1+k) + \max\{\log\frac{\delta_0}{d}, 0\}\right)$, $\eta_0 = d^{-2/3}$, $\delta_0 = d^{1/3}$, $\alpha \in (0,1)$, $\beta \in (0, \frac{1}{2})$. If $2\alpha - 4\beta < 1$, then there exists a constant $t_0$ such that the iterates $\{\boldsymbol{\theta}_k\}_{k\geq 0}$ satisfies*

$$\frac{1}{1+T}\sum_{k=0}^{T}\mathbb{E}\|\nabla\mathcal{L}(\boldsymbol{\theta}_k)\|^2 \leq 12\max\{c_5(1-\lambda), c_6, \frac{c_7}{1-\lambda}\}d^{2/3}T^{-\min\{2\beta, 1-\alpha, \alpha-2\beta\}}, \quad \forall T \geq t_0.$$

Optimizing the step size exponents $\alpha, \beta$ in the above concludes the proof of Theorem 3.1.

### 4.1 Discussions

We conclude by discussing two alternative zero-th order gradient estimators to (4), and argue that they do not improve over the sample complexity in the proposed $\mathtt{DFO}(\lambda)$ algorithm. We study:

$$\boldsymbol{g}_{\mathsf{2pt-I}} := \tfrac{d}{\delta} \left[ \ell \left( \boldsymbol{\theta} + \delta \boldsymbol{u}; Z \right) - \ell(\boldsymbol{\theta}; Z) \right] \boldsymbol{u}, \quad \boldsymbol{g}_{\mathsf{2pt-II}} := \tfrac{d}{\delta} \left[ \ell \left( \boldsymbol{\theta} + \delta \boldsymbol{u}; Z_1 \right) - \ell(\boldsymbol{\theta}; Z_2) \right] \boldsymbol{u}, \quad (20)$$

where $\boldsymbol{u} \sim \mathsf{Unif}(\mathbb{S}^{\mathsf{d}-1})$. For ease of illustration, we assume that the samples $Z, Z_1, Z_2$ are drawn directly from the stationary distributions $Z \sim \Pi_{\boldsymbol{\theta}+\delta u}, Z_1 \sim \Pi_{\boldsymbol{\theta}+\delta u}, Z_2 \sim \Pi_{\boldsymbol{\theta}}$.

We recall from §2 that the estimator $\boldsymbol{g}_{\mathsf{2pt-I}}$ is a finite difference approximation of the directional derivative of objective function along the randomized direction $\boldsymbol{u}$[1], as proposed in Nesterov & Spokoiny (2017); Ghadimi & Lan (2013). For non-convex stochastic optimization with decision independent sample distribution, i.e., $\Pi_{\boldsymbol{\theta}} \equiv \bar{\Pi}$ for all $\boldsymbol{\theta}$, the DFO algorithm based on $\boldsymbol{g}_{\mathsf{2pt-I}}$ is known to admit an optimal sample complexity of $\mathcal{O}(1/\epsilon^2)$ (Jamieson et al., 2012). Note that $\mathbb{E}_{\boldsymbol{u}\sim\mathsf{Unif}(\mathbb{S}^{\mathsf{d}-1}),Z\sim\bar{\Pi}}[\ell(\boldsymbol{\theta};Z)\boldsymbol{u}] = \boldsymbol{0}$. However, in the case of decision-dependent sample distribution as in (1), $\boldsymbol{g}_{\mathsf{2pt-I}}$ would become a *biased* estimator since the sample $Z$ is drawn from $\Pi_{\boldsymbol{\theta}+\delta\boldsymbol{u}}$ which depends on $\boldsymbol{u}$. The DFO algorithm based on $\boldsymbol{g}_{\mathsf{2pt-I}}$ may not converge to a stationary solution of (1).

A remedy to handle the above issues is to consider the estimator $\boldsymbol{g}_{\mathsf{2pt-II}}$ which utilizes *two samples* $Z_1, Z_2$, each independently drawn at a different decision variable, to form the gradient estimate. In fact, it can be shown that $\mathbb{E}[\boldsymbol{g}_{\mathsf{2pt-II}}] = \nabla \mathcal{L}_{\delta}(\boldsymbol{\theta})$ yields an unbiased gradient estimator. However, due to the decoupled random samples $Z_1, Z_2$, we have

$$\mathbb{E} \left\| \boldsymbol{g}_{\mathsf{2pt-II}} \right\|^2 = \mathbb{E} \left[ \left( \ell \left( \boldsymbol{\theta} + \delta \boldsymbol{u}; Z_1 \right) - \ell(\boldsymbol{\theta}; Z_1) + \ell(\boldsymbol{\theta}; Z_1) - \ell(\boldsymbol{\theta}; Z_2) \right)^2 \right] \frac{d^2}{\delta^2}$$

$$\overset{(a)}{\geq} \mathbb{E} \left[ \frac{3}{4} \left( \ell(\boldsymbol{\theta}; Z_1) - \ell(\boldsymbol{\theta}; Z_2) \right)^2 - 3 \left( \ell \left( \boldsymbol{\theta} + \delta \boldsymbol{u}; Z_1 \right) - \ell(\boldsymbol{\theta}; Z_1) \right)^2 \right] \frac{d^2}{\delta^2}$$

$$= \frac{3}{2} \mathsf{Var}[\ell(\boldsymbol{\theta}; Z)] \frac{d^2}{\delta^2} - 3\mathbb{E} \left[ \left( \ell \left( \boldsymbol{\theta} + \delta \boldsymbol{u}; Z_1 \right) - \ell(\boldsymbol{\theta}; Z_1) \right)^2 \right] \frac{d^2}{\delta^2} \overset{(b)}{\geq} \frac{3}{2} \frac{\sigma^2 d^2}{\delta^2} - 3\mu^2 d^2 = \Omega(1/\delta^2).$$

where in (a) we use the fact that $(x + y)^2 \geq \frac{3}{4}x^2 - 3y^2$, in (b) we assume $\mathsf{Var}[\ell(\boldsymbol{\theta}; Z)] := \mathbb{E}\left( \ell(\boldsymbol{\theta}; Z) - \mathcal{L}(\boldsymbol{\theta}) \right)^2 \geq \sigma^2 > 0$ and $\ell(\boldsymbol{\theta}; z)$ is $\mu$-Lipschitz in $\boldsymbol{\theta}$. As such, this two-point gradient estimator does not reduce the variance when compared with the estimator in (4). Note that a two-sample estimator also incurs additional sampling overhead in the scenario of Markovian samples.

## 5 Numerical Experiments

We examine the efficacy of the $\mathtt{DFO}(\lambda)$ algorithm on a few toy examples by comparing $\mathtt{DFO}(\lambda)$ with a simple stochastic gradient descent scheme with greedy deployment. Unless otherwise specified, we use the step size choices in (8) for $\mathtt{DFO}(\lambda)$. All experiments are conducted on a server with an Intel Xeon 6318 CPU using Python 3.7. To measure performance, we record the gradient norm $\|\nabla\mathcal{L}(\boldsymbol{\theta})\|$ and estimate its expected value using at least 8 trials.

**1-Dimensional Case: Quadratic Loss.** The first example considers a scalar quadratic loss function $\ell : \mathbb{R} \times \mathbb{R} \to \mathbb{R}$ defined by $\ell(\boldsymbol{\theta}; z) = \frac{1}{12} z\boldsymbol{\theta}(3\boldsymbol{\theta}^2 - 8\boldsymbol{\theta} - 48)$. To simulate the controlled Markov chain scenario, the samples are generated dynamically according to an auto-regressive (AR) process $Z_{t+1} = (1 - \gamma)Z_t + \gamma\bar{Z}_{t+1}$ with $\bar{Z}_{t+1} \sim \mathcal{N}(\boldsymbol{\theta}, \frac{(2-\gamma)}{\gamma}\sigma^2)$ with parameter $\gamma \in (0, 1)$. Note that the stationary distribution of the AR process is $\Pi_{\boldsymbol{\theta}} = \mathcal{N}(\boldsymbol{\theta}, \sigma^2)$. As such, the performative risk function in this case is $\mathcal{L}(\boldsymbol{\theta}) = \mathbb{E}_{Z\sim\Pi_{\boldsymbol{\theta}}}[\ell(\boldsymbol{\theta}; Z)] = \frac{\boldsymbol{\theta}^2}{12}(\boldsymbol{\theta}^2 - 8\boldsymbol{\theta} - 48)$, which is quartic in $\boldsymbol{\theta}$. Note that $\mathcal{L}(\boldsymbol{\theta})$ is not convex in $\boldsymbol{\theta}$ and the set of stationary solution is $\{\boldsymbol{\theta} : \nabla\mathcal{L}(\boldsymbol{\theta}) = 0\} = \{4, 0, -2\}$, among which the optimal solution is $\boldsymbol{\theta}_{PO} = \arg\min_{\boldsymbol{\theta}} \mathcal{L}(\boldsymbol{\theta}) = 4$.

In our experiments below, we initialize all the algorithms are initialized by $\boldsymbol{\theta}_0 = 6$. In Figure 1 (left), we compare the norms of the gradient for performative risk with pure $\mathtt{DFO}$ (no burn-in), the $\mathtt{DFO}(\lambda)$ algorithm, and stochastic gradient descent with greedy deployment scheme (SGD-GD) against the number of samples observed by the algorithms. We first observe from Figure 1 (left) that pure $\mathtt{DFO}$ and $\mathtt{SGD-GD}$ methods do not converge to a stationary point to $\mathcal{L}(\boldsymbol{\theta})$ even after more samples

---

[1]Note that in Nesterov & Spokoiny (2017); Ghadimi & Lan (2013), the random vector $\boldsymbol{u}$ is drawn from a Gaussian distribution.

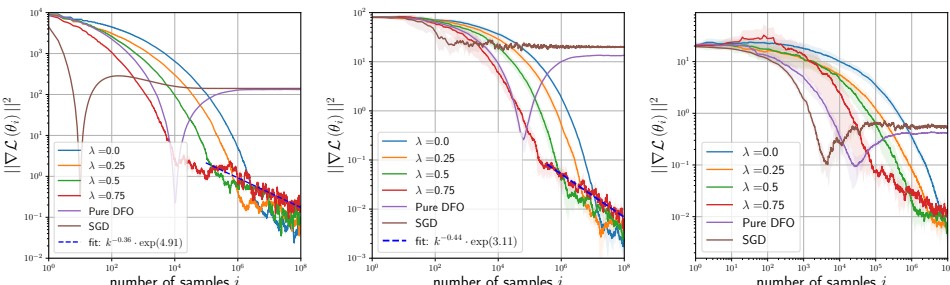

Figure 1: (*left*) One Dimension Quadratic Minimization problem with samples generated by AR distribution model where regressive parameter $\gamma = 0.5$. (*middle*) Markovian Pricing Problem with $d = 5$ dimension. (*right*) Linear Regression problem based on AR distribution model ($\gamma = 0.5$).

are observed. On the other hand, DFO $(\lambda)$ converges to a stationary point of $\mathcal{L}(\boldsymbol{\theta})$ at the rate of $\|\nabla\mathcal{L}(\boldsymbol{\theta})\|^2 = \mathcal{O}(1/S^{0.36})$, matching Theorem 3.1 that predicts a rate of $\mathcal{O}(1/S^{1/3})$, where $S$ is the total number of samples observed.

Besides, we observe that with large $\lambda = 0.75$, DFO $(\lambda)$ converges at a faster rate at the beginning (i.e., transient phase), but the convergence rate slows down at the steady phase (e.g., when no. of samples observed is greater than $10^6$) compared to running the same algorithm with smaller $\lambda$.

**Higher Dimension Case: Markovian Pricing.** The second example examines a multi-dimensional ($d = 5$) pricing problem similar to (Izzo et al., 2021, Sec. 5.2). The decision variable $\boldsymbol{\theta} \in \mathbb{R}^5$ denotes the prices of $d = 5$ goods and $\kappa$ is a drifting parameter for the prices. Our goal is to maximize the average revenue $\mathbb{E}_{Z\sim\Pi_{\boldsymbol{\theta}}}[\ell(\boldsymbol{\theta}; Z)]$ with $\ell(\boldsymbol{\theta}; z) = -\langle\boldsymbol{\theta}\,|\,z\rangle$, where $\Pi_{\boldsymbol{\theta}} \equiv \mathcal{N}(\boldsymbol{\mu}_0 - \kappa\boldsymbol{\theta}, \sigma^2\boldsymbol{I})$ is the unique stationary distribution of the Markov process (i.e., an AR process)

$$Z_{t+1} = (1-\gamma)Z_t + \gamma\bar{Z}_{t+1} \ \text{ with } \ \bar{Z}_{t+1} \sim \mathcal{N}(\boldsymbol{\mu}_0 - \kappa\boldsymbol{\theta}, \tfrac{2-\gamma}{\gamma}\sigma^2\boldsymbol{I}).$$

Note that in this case, the performative optimal solution is $\boldsymbol{\theta}_{PO} = \arg\min_{\boldsymbol{\theta}}\mathcal{L}(\boldsymbol{\theta}) = \boldsymbol{\mu}_0/(2\kappa)$.

We set $\gamma = 0.5, \sigma = 5$, drifting parameter $\kappa = 0.5$, initial mean of non-shifted distribution $\boldsymbol{\mu}_0 = [-2, 2, -2, 2, -2]^\top$. All the algorithms are initialized by $\boldsymbol{\theta}_0 = [2, -2, 2, -2, 2]^\top$. We simulate the convergence behavior for different algorithms in Figure 1 (middle). Observe that the differences between the DFO $(\lambda)$ algorithms with different $\lambda$ becomes less significant than Figure 1 (left).

**Markovian Performative Regression.** The last example considers the linear regression problem in (Nagaraj et al., 2020) which is a prototype problem for studying stochastic optimization with Markovian data (e.g., reinforcement learning). Unlike the previous examples, this problem involves a pair of correlated r.v.s that follows a decision-dependent joint distribution. We adopt a setting similar to the regression example in (Izzo et al., 2021), where $(X, Y) \sim \Pi_{\boldsymbol{\theta}}$ with $X \sim \mathcal{N}(0, \sigma_1^2\boldsymbol{I}), Y|X \sim \mathcal{N}\left(\langle\beta(\boldsymbol{\theta})\,|\,X\rangle, \sigma_2^2\right), \beta(\boldsymbol{\theta}) = \boldsymbol{a}_0 + a_1\boldsymbol{\theta}$. The loss function is $\ell(\boldsymbol{\theta}; x, y) = (\langle x\,|\,\boldsymbol{\theta}\rangle - y)^2 + \frac{\mu}{2}\|\boldsymbol{\theta}\|^2$. In this case, the performative risk is:

$$\mathcal{L}(\boldsymbol{\theta}) = \mathbb{E}_{\Pi_{\boldsymbol{\theta}}}\left[\ell(\boldsymbol{\theta}; X, Y)\right] = (\sigma_1^2 a_1^2 - 2\sigma_1^2 a_1 + \sigma_1^2 + \tfrac{\mu}{2})\|\boldsymbol{\theta}\|^2 - 2\sigma_1^2(1-a_1)\boldsymbol{\theta}^\top\boldsymbol{a}_0 + \sigma_1^2\|\boldsymbol{a}_0\|^2 + \sigma_2^2,$$

For simplicity, we assume $\sigma_1^2(1-a_1) = \sigma_1^2 a_1^2 - 2\sigma_1^2 a_1 + \sigma_1^2 + \mu/2$, from which we can deduce $\boldsymbol{\theta}_{PO} = \boldsymbol{a}_0$. In this experiment, we consider Markovian samples $(\tilde{X}_t, \tilde{Y}_t)_{t=1}^T$ drawn from an AR process:

$$(\tilde{X}_t, \tilde{Y}_t) = (1-\gamma)(\tilde{X}_{t-1}, \tilde{Y}_{t-1}) + \gamma(X_t, Y_t),$$
$$X_t \sim \mathcal{N}(0, \tfrac{2-\gamma}{\gamma}\sigma_1^2 I), \ Y_t|X_t \sim \mathcal{N}(\langle\tilde{X}_t\,|\,\beta(\boldsymbol{\theta}_{t-1})\rangle, \tfrac{2-\gamma}{\gamma}\sigma_2^2),$$

for any $t \geq 1$. We set $d = 5, \boldsymbol{a}_0 = [-1, 1, -1, 1, -1]^\top, a_1 = 0.5, \sigma_1^2 = \sigma_2^2 = 1$, regularization parameter $\mu = 0.5$, mixing parameter $\gamma = 0.1$. The algorithms are initialized with $\boldsymbol{\theta}_0 = [1, -1, 1, -1, 1]^\top$. Figure 1 (right) shows the result of the simulation. Similar to the previous examples, we observe that pure DFO and SGD fail to find a stationary solution to $\mathcal{L}(\boldsymbol{\theta})$. Meanwhile, DFO $(\lambda)$ converges to a stationary solution after a reasonable number of samples are observed.

**Conclusions.** We have described a derivative-free optimization approach for finding a stationary point of the performative risk function. In particular, we consider a non-i.i.d. data setting with samples generated from a controlled Markov chain and propose a two-timescale step sizes approach in constructing the gradient estimator. The proposed DFO $(\lambda)$ algorithm is shown to converge to a stationary point of the performative risk function at the rate of $\mathcal{O}(1/T^{1/3})$.

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
