# A  Proof of Lemma 4.1

*Proof.* Throughout this section, we let $\check{\boldsymbol{\theta}}_k := \boldsymbol{\theta}_k + \delta_k u_k, g_k(\boldsymbol{\theta}; u, z) := g_{\delta_k}(\boldsymbol{\theta}; u, z)$ and $\mathcal{L}_k(\boldsymbol{\theta}) := \mathcal{L}_{\delta_k}(\boldsymbol{\theta})$ for simplicity. We begin our analysis from Assumption 3.1 and the observation that $\boldsymbol{\theta}_{k+1} - \boldsymbol{\theta}_k = -\eta_k \sum_{m=1}^{\tau_k} \lambda^{\tau_k - m} g_k^{(m)}$. Recall that $g_k^{(m)} = \frac{d}{\delta_k} \ell\left(\check{\boldsymbol{\theta}}_k^{(m)}; z_k^{(m)}\right) u_k$ and $\check{\boldsymbol{\theta}}_k^{(m)} = \boldsymbol{\theta}_k^{(m)} + \delta_k u_k$, we have

$$\mathcal{L}(\boldsymbol{\theta}_{k+1}) - \mathcal{L}(\boldsymbol{\theta}_k) + \eta_k \left\langle \nabla \mathcal{L}(\boldsymbol{\theta}_k) \,\middle|\, \sum_{m=1}^{\tau_k} \lambda^{\tau_k - m} g_k^{(m)} \right\rangle \leq \frac{L}{2} \eta_k^2 \left\| \sum_{m=1}^{\tau_k} \lambda^{\tau_k - m} g_k^{(m)} \right\|^2,$$

Rearranging terms and adding $\frac{\eta_k}{1-\lambda} \|\nabla \mathcal{L}(\boldsymbol{\theta}_k)\|^2$ on the both sides lead to

$$\frac{\eta_k}{1-\lambda} \|\nabla \mathcal{L}(\boldsymbol{\theta}_k)\|^2 \leq \mathcal{L}(\boldsymbol{\theta}_k) - \mathcal{L}(\boldsymbol{\theta}_{k+1}) - \frac{\eta_k}{1-\lambda} \left\langle \nabla \mathcal{L}(\boldsymbol{\theta}_k) \,\middle|\, (1-\lambda) \sum_{m=1}^{\tau_k} \lambda^{\tau_k - m} g_k^{(m)} - \nabla \mathcal{L}(\boldsymbol{\theta}_k) \right\rangle$$
$$+ \frac{L}{2} \eta_k^2 \left\| \sum_{m=1}^{\tau_k} \lambda^{\tau_k - m} g_k^{(m)} \right\|^2$$

Let $\mathcal{F}^k = \sigma(\boldsymbol{\theta}_0, Z_s^{(m)}, u_s, 0 \leq s \leq k, 0 \leq m \leq \tau_k)$ be the filtration of random variables. Taking expectation conditioned on $\mathcal{F}^{k-1}$ gives

$$\frac{\eta_k}{1-\lambda} \|\nabla \mathcal{L}(\boldsymbol{\theta}_k)\|^2 \leq \mathbb{E}_{\mathcal{F}^{k-1}} [\mathcal{L}(\boldsymbol{\theta}_k) - \mathcal{L}(\boldsymbol{\theta}_{k+1})]$$
$$- \frac{\eta_k}{1-\lambda} \left\langle \nabla \mathcal{L}(\boldsymbol{\theta}_k) \,\middle|\, (1-\lambda) \sum_{m=1}^{\tau_k} \lambda^{\tau_k - m} \mathbb{E}_{\mathcal{F}^{k-1}}\left[g_k^{(m)}\right] - \nabla \mathcal{L}(\boldsymbol{\theta}_k) \right\rangle$$
$$+ \frac{L}{2} \eta_k^2 \mathbb{E}_{\mathcal{F}^{k-1}} \left\| \sum_{m=1}^{\tau_k} \lambda^{\tau_k - m} g_k^{(m)} \right\|^2,$$

By adding and subtracting, we obtain

$$\frac{\eta_k}{1-\lambda} \|\nabla \mathcal{L}(\boldsymbol{\theta}_k)\|^2 \leq \mathbb{E}_{\mathcal{F}^{k-1}} [\mathcal{L}(\boldsymbol{\theta}_k) - \mathcal{L}(\boldsymbol{\theta}_{k+1})]$$
$$- \frac{\eta_k}{1-\lambda} \left\langle \nabla \mathcal{L}(\boldsymbol{\theta}_k) \,\middle|\, (1-\lambda) \sum_{m=1}^{\tau_k} \lambda^{\tau_k - m} \left( \mathbb{E}_{\mathcal{F}^{k-1}}\left[g_k^{(m)}\right] - \mathbb{E}_{Z \sim \Pi_{\check{\boldsymbol{\theta}}_k}, \mathcal{F}^{k-1}}[g_k(\boldsymbol{\theta}_k; u_k, Z)] \right) \right\rangle$$
$$- \frac{\eta_k}{1-\lambda} \left\langle \nabla \mathcal{L}(\boldsymbol{\theta}_k) \,\middle|\, (1-\lambda) \sum_{m=1}^{\tau_k} \lambda^{\tau_k - m} \mathbb{E}_{Z \sim \Pi_{\check{\boldsymbol{\theta}}_k}, \mathcal{F}^{k-1}}[g_k(\boldsymbol{\theta}_k; u_k, Z)] - \nabla \mathcal{L}(\boldsymbol{\theta}_k) \right\rangle$$
$$+ \frac{L}{2} \eta_k^2 \mathbb{E}_{\mathcal{F}^{k-1}} \left\| \sum_{m=1}^{\tau_k} \lambda^{\tau_k - m} g_k^{(m)} \right\|^2$$

By Lemma E.2, the conditional expectation evaluates to $\mathbb{E}_{Z \sim \Pi_{\check{\boldsymbol{\theta}}_k}}[g_k(\boldsymbol{\theta}_k; u_k, Z)] = \nabla \mathcal{L}_k(\boldsymbol{\theta}_k)$. Dividing $\frac{\eta_k}{1-\lambda}$ derive that

$$\|\nabla \mathcal{L}(\boldsymbol{\theta}_k)\|^2 \leq \frac{1-\lambda}{\eta_k} \mathbb{E}_{\mathcal{F}^{k-1}} (\mathcal{L}(\boldsymbol{\theta}_k) - \mathcal{L}(\boldsymbol{\theta}_{k+1}))$$
$$- \left\langle \nabla \mathcal{L}(\boldsymbol{\theta}_k) \,\middle|\, (1-\lambda) \sum_{m=1}^{\tau_k} \lambda^{\tau_k - m} \left( \mathbb{E}_{\mathcal{F}^{k-1}}\left[g_k^{(m)}\right] - \mathbb{E}_{\mathcal{F}^{k-1}} \mathbb{E}_{Z \sim \Pi_{\check{\boldsymbol{\theta}}_k}}[g_k(\boldsymbol{\theta}_k; u_k, Z)|u_k] \right) \right\rangle$$
$$- \left\langle \nabla \mathcal{L}(\boldsymbol{\theta}_k) \,\middle|\, (1-\lambda) \left( \sum_{m=1}^{\tau_k} \lambda^{\tau_k - m} \nabla \mathcal{L}_k(\boldsymbol{\theta}_k) \right) - \nabla \mathcal{L}(\boldsymbol{\theta}_k) \right\rangle$$
$$+ \frac{L(1-\lambda)}{2} \eta_k \mathbb{E}_{\mathcal{F}^{k-1}} \left\| \sum_{m=1}^{\tau_k} \lambda^{\tau_k - m} g_k^{(m)} \right\|^2$$

Summing over $k$ from 0 to $t$, indeed we obtain

$$\sum_{k=0}^{t} \mathbb{E}\left\|\nabla\mathcal{L}(\boldsymbol{\theta}_k)\right\|^2$$

$$\leq \sum_{k=0}^{t} \frac{1-\lambda}{\eta_k}\mathbb{E}\left[\mathcal{L}(\boldsymbol{\theta}_k) - \mathcal{L}(\boldsymbol{\theta}_{k+1})\right]$$

$$- \sum_{k=0}^{t}\mathbb{E}\left\langle \nabla\mathcal{L}(\boldsymbol{\theta}_k) \,\middle|\, (1-\lambda)\sum_{m=1}^{\tau_k}\lambda^{\tau_k-m}\left(\mathbb{E}_{\mathcal{F}^{k-1}}\left[g_k^{(m)}\right] - \mathbb{E}_{\mathcal{F}^{k-1}}\mathbb{E}_{Z\sim\Pi_{\hat{\boldsymbol{\theta}}_k}}\left[g_k(\boldsymbol{\theta}_k;u_k,Z)|u_k\right]\right)\right\rangle$$

$$- \sum_{k=0}^{t}\mathbb{E}\left\langle \nabla\mathcal{L}(\boldsymbol{\theta}_k) \,\middle|\, (1-\lambda)\left(\sum_{m=1}^{\tau_k}\lambda^{\tau_k-m}\nabla\mathcal{L}_k(\boldsymbol{\theta}_k)\right) - \nabla\mathcal{L}(\boldsymbol{\theta}_k)\right\rangle$$

$$+ \frac{L(1-\lambda)}{2}\sum_{k=0}^{t}\eta_k\mathbb{E}\left\|\sum_{m=1}^{\tau_k}\lambda^{\tau_k-m}g_k^{(m)}\right\|^2 := \mathbf{I}_1(t) + \mathbf{I}_2(t) + \mathbf{I}_3(t) + \mathbf{I}_4(t)$$

$\square$

## B  Proof of Lemma 4.2

**Lemma B.1.** *Under Assumption 3.2 and step size $\eta_t = \eta_0(1+t)^{-\alpha}$, it holds that*

$$\mathbf{I}_1(t) \leq c_1(1-\lambda)(1+t)^{\alpha} \tag{21}$$

*where constant $c_1 = \frac{2G}{\eta_0}$.*

*Proof.* We observe the following chain

$$\mathbf{I}_1(t) = \sum_{k=0}^{t}\frac{1-\lambda}{\eta_k}\left(\mathbb{E}\left[\mathcal{L}(\boldsymbol{\theta}_k)\right] - \mathbb{E}\left[\mathcal{L}(\boldsymbol{\theta}_{k+1})\right]\right)$$

$$= (1-\lambda)\sum_{k=0}^{t}\mathbb{E}[\mathcal{L}(\boldsymbol{\theta}_k)]/\eta_k - \mathbb{E}[\mathcal{L}(\boldsymbol{\theta}_{k+1})]/\eta_{k+1} + \mathbb{E}[\mathcal{L}(\boldsymbol{\theta}_{k+1})]/\eta_{k+1} - \mathbb{E}[\mathcal{L}(\boldsymbol{\theta}_{k+1})]/\eta_k$$

$$\overset{(a)}{=} (1-\lambda)\left[\mathbb{E}[\mathcal{L}(\boldsymbol{\theta}_0)/\eta_0] - \mathbb{E}[\mathcal{L}(\boldsymbol{\theta}_{t+1})/\eta_{t+1}] + \sum_{k=0}^{t}(\frac{1}{\eta_{k+1}} - \frac{1}{\eta_k})\mathbb{E}[\mathcal{L}(\boldsymbol{\theta}_{k+1})]\right]$$

$$\leq (1-\lambda)\max_{k}|\mathbb{E}[\mathcal{L}(\boldsymbol{\theta}_k)]|\left(\frac{1}{\eta_0} + \frac{1}{\eta_{t+1}} + \frac{1}{\eta_{t+1}} - \frac{1}{\eta_0}\right)$$

where equality (a) is obtained using the fact that step size $\eta_k > 0$ is a decreasing sequence. Applying assumption 3.2 to the last inequality leads to

$$\mathbf{I}_1(t) \leq (1-\lambda)G\frac{2}{\eta_{t+1}}$$

$$\leq c_1(1-\lambda)(1+t)^{\alpha}$$

where the constant $c_1 = \frac{2G}{\eta_0}$.

$\square$

**Lemma B.2.** *Under Assumption 3.1, 3.2, 3.3, 3.4, 3.5, and constraint $0 < 2\alpha - 4\beta < 1$, and for all $k \geq 0$, $\tau_k \geq \frac{1}{\log 1/\max\{\rho,\lambda\}}\log(1+k)$, then there exists universal constants $t_1, t_2 > 0$ such that*

$$\mathbf{I}_2(t) \leq c_2\frac{d^2}{(1-\lambda)^2}\mathcal{A}(t)^{1/2}(1+t)^{1-(\alpha-2\beta)} \quad \forall t \geq \max\{t_1, t_2\} \tag{22}$$

*where $\mathcal{A}(t) := \frac{1}{1+t}\sum_{k=0}^{t}\mathbb{E}\left\|\nabla\mathcal{L}(\boldsymbol{\theta}_k)\right\|^2$ and $c_2 := \frac{\eta_0}{\delta_0^2}\frac{6\cdot(L_1G^2+L_2G^2+\sqrt{L}G^{3/2})}{\sqrt{1-2\alpha+4\beta}}$ is a constant.*

*Proof.* Fix $k > 0$, and recall $\check{\boldsymbol{\theta}}_k := \boldsymbol{\theta}_k + \delta_k u_k, \check{\boldsymbol{\theta}}_k^{(\ell)} := \boldsymbol{\theta}_k^{(\ell)} + \delta_k u_k$, then consider the following pair of Markov chains:

$$Z_k = Z_k^{(0)} \xrightarrow{\check{\boldsymbol{\theta}}_k^{(1)}} Z_k^{(1)} \xrightarrow{\check{\boldsymbol{\theta}}_k^{(2)}} Z_k^{(2)} \xrightarrow{\check{\boldsymbol{\theta}}_k^{(3)}} Z_k^{(3)} \cdots \xrightarrow{\check{\boldsymbol{\theta}}_k^{(\tau_k)}} Z_k^{(\tau_k)} = Z_{k+1} \tag{23}$$

$$Z_k = \tilde{Z}_k^{(0)} \xrightarrow{\check{\boldsymbol{\theta}}_k} \tilde{Z}_k^{(1)} \xrightarrow{\check{\boldsymbol{\theta}}_k} \tilde{Z}_k^{(2)} \xrightarrow{\check{\boldsymbol{\theta}}_k} \tilde{Z}_k^{(3)} \cdots \xrightarrow{\check{\boldsymbol{\theta}}_k} \tilde{Z}_k^{(\tau_k)} \tag{24}$$

where the arrow associated with $\boldsymbol{\theta}$ represents the transition kernel $\mathbb{T}_{\boldsymbol{\theta}}(\cdot, \cdot)$.

Note that Chain 23 is the trajectory of $\mathsf{DFO}(\lambda)$ algorithm at iteration $k$, while Chain 24 describes the trajectory of the same length generated by a reference Markov chain with fixed transition kernel $\mathbb{T}_{\check{\boldsymbol{\theta}}_k}(\cdot, \cdot)$. Since $Z_k = Z_k^{(0)} = \tilde{Z}_k^{(0)}$, we shall use them interchangeably.

Define $\Delta_{k,m} := \mathbb{E}_{\mathcal{F}^{k-1}} \left[ g_k^{(m)} - \mathbb{E}_{Z \sim \Pi_{\check{\boldsymbol{\theta}}_k}} [g_k(\boldsymbol{\theta}_k; u_k, Z)] \right]$, then $\mathbf{I}_2(t)$ can be reformed as

$$\mathbf{I}_2(t) = -(1 - \lambda) \mathbb{E} \sum_{k=0}^{t} \left\langle \nabla \mathcal{L}(\boldsymbol{\theta}_k) \mid \sum_{m=1}^{\tau_k} \lambda^{\tau_k - m} \Delta_{k,m} \right\rangle$$

$$\leq (1 - \lambda) \mathbb{E} \sum_{k=0}^{t} \|\nabla \mathcal{L}(\boldsymbol{\theta}_k)\| \cdot \left\| \sum_{m=1}^{\tau_k} \lambda^{\tau_k - m} \Delta_{k,m} \right\|$$

Next, observe that each $\Delta_{k,m}$ can be decomposed into 3 bias terms as follows

$$\Delta_{k,m} = \mathbb{E}_{\mathcal{F}^{k-1}} \left[ \frac{d}{\delta_k} \left( \mathbb{E}[\ell(\check{\boldsymbol{\theta}}_k^{(m)}; Z_k^{(m)}) | \check{\boldsymbol{\theta}}_k^{(m)}, Z_k^{(0)}] - \mathbb{E}_{Z \sim \Pi_{\check{\boldsymbol{\theta}}_k}} [\ell(\check{\boldsymbol{\theta}}_k; Z) | \check{\boldsymbol{\theta}}_k] \right) u_k \right]$$

$$= \mathbb{E}_{\mathcal{F}^{k-1}} \left[ \frac{d}{\delta_k} \left( \mathbb{E}[\ell(\check{\boldsymbol{\theta}}_k^{(m)}; Z_k^{(m)}) | \check{\boldsymbol{\theta}}_k^{(m)}, Z_k^{(0)}] - \mathbb{E}_{\tilde{Z}_k^{(m)}} [\ell(\check{\boldsymbol{\theta}}_k^{(m)}; \tilde{Z}_k^{(m)}) | \check{\boldsymbol{\theta}}_k^{(m)}, \tilde{Z}_k^{(0)}] \right) u_k \right]$$

$$+ \mathbb{E}_{\mathcal{F}^{k-1}} \left[ \frac{d}{\delta_k} \left( \mathbb{E}_{\tilde{Z}_k^{(m)}} [\ell(\check{\boldsymbol{\theta}}_k^{(m)}; \tilde{Z}_k^{(m)}) | \check{\boldsymbol{\theta}}_k^{(m)}, \tilde{Z}_k^{(0)}] - \mathbb{E}_{Z \sim \Pi_{\check{\boldsymbol{\theta}}_k}} [\ell(\check{\boldsymbol{\theta}}_k^{(m)}; Z) | \check{\boldsymbol{\theta}}_k^{(m)}] \right) u_k \right]$$

$$+ \mathbb{E}_{\mathcal{F}^{k-1}} \frac{d}{\delta_k} \underbrace{\mathbb{E}_{Z \sim \Pi_{\check{\boldsymbol{\theta}}_k}} \left[ \ell(\check{\boldsymbol{\theta}}_k^{(m)}; Z) - \ell(\check{\boldsymbol{\theta}}_k; Z) | \check{\boldsymbol{\theta}}_k^{(m)}, \check{\boldsymbol{\theta}}_k \right]}_{\leq c_8 \left\| \check{\boldsymbol{\theta}}_k^{(m)} - \check{\boldsymbol{\theta}}_k \right\| + \frac{L}{2} \left\| \check{\boldsymbol{\theta}}_k^{(m)} - \check{\boldsymbol{\theta}}_k \right\|^2} u_k$$

where we use Lemma E.3 in the last inequality and $c_8 := 2 \left( \sqrt{LG} + GL_1 \right)$.

Here we bound these three parts separately. For the first term, it holds that

$$\left| \mathbb{E}[\ell(\check{\boldsymbol{\theta}}_k^{(m)}; Z_k^{(m)}) | \check{\boldsymbol{\theta}}_k^{(m)}, Z_k^{(0)}] - \mathbb{E}_{\tilde{Z}_k^{(m)}} [\ell(\check{\boldsymbol{\theta}}_k^{(m)}; \tilde{Z}_k^{(m)}) | \check{\boldsymbol{\theta}}_k^{(m)}, \tilde{Z}_k^{(0)}] \right|$$

$$= \left| \int_{\mathsf{Z}} \ell(\check{\boldsymbol{\theta}}_k^{(m)}; z) \mathbb{P}(Z_k^{(m)} = z | Z_k^{(0)}) - \ell(\check{\boldsymbol{\theta}}_k^{(m)}; z) \mathbb{P}(\tilde{Z}_k^{(m)} = z | \tilde{Z}_k^{(0)}) dz \right|$$

$$\leq G \int_{\mathsf{Z}} \left| \mathbb{P}(Z_k^{(m)} = z | Z_k^{(0)}) - \mathbb{P}(\tilde{Z}_k^{(m)} = z | \tilde{Z}_k^{(0)}) \right| dz$$

$$= 2G \boldsymbol{\delta}_{\mathrm{TV}} \left( \mathbb{P}(z_k^{(m)} \in \cdot | Z_k^{(0)}), \mathbb{P}(\tilde{Z}_k^{(m)} \in \cdot | Z_k^{(0)}) \right)$$

$$\leq 2GL_2 \sum_{\ell=1}^{m-1} \left\| \check{\boldsymbol{\theta}}_k^{(\ell)} - \check{\boldsymbol{\theta}}_k \right\| = 2GL_2 \sum_{\ell=1}^{m-1} \left\| \boldsymbol{\theta}_k^{(\ell)} - \boldsymbol{\theta}_k \right\|$$

where the first inequality is due to Assumption 3.2, the second inequality is due to Lemma E.4. For the second term, we have

$$\left| \mathbb{E}_{\tilde{Z}_k^{(m)}} [\ell(\check{\boldsymbol{\theta}}_k^{(m)}; Z_k^{(m)})] - \mathbb{E}_{Z \sim \Pi_{\check{\boldsymbol{\theta}}_k}} [\ell(\check{\boldsymbol{\theta}}_k; Z)] \right|$$

$$= \left| \int_{\mathsf{Z}} \ell(\check{\boldsymbol{\theta}}_k^{(m)}; z) \mathbb{P}(\tilde{Z}_k^{(m)} = z | \tilde{Z}_k^{(0)}) - \ell(\check{\boldsymbol{\theta}}_k^{(m)}; z) \Pi_{\check{\boldsymbol{\theta}}_k}(z) dz \right|$$

$$\overset{(a)}{\leq} G \int_{\mathsf{Z}} |\mathbb{P}(\tilde{Z}_k^{(m)} = z | \tilde{Z}_k^{(0)}) - \Pi_{\check{\boldsymbol{\theta}}_k}(z)) | dz$$

$$=2G\boldsymbol{\delta}_{\mathrm{TV}}\left(\mathbb{P}(\tilde{Z}_k^{(m)}\in\cdot|\tilde{Z}_k^{(0)}),\Pi_{\check{\boldsymbol{\theta}}_k}\right)$$

$$\leq 2GM\rho^m$$

where we use Assumption 3.2 in inequality (a) and Assumptions 3.4 in the last inequality. Combining three upper bounds, we obtain that

$$\|\Delta_{k,m}\|\leq\mathbb{E}_{\mathcal{F}^{k-1}}\frac{d}{\delta_k}\left(2GL_2\sum_{\ell=1}^{m-1}\left[\left\|\boldsymbol{\theta}_k^{(\ell)}-\boldsymbol{\theta}_k\right\|\right]+2GM\rho^m+c_8\left\|\check{\boldsymbol{\theta}}_k^{(m)}-\check{\boldsymbol{\theta}}_k\right\|+\frac{L}{2}\left\|\check{\boldsymbol{\theta}}_k^{(m)}-\check{\boldsymbol{\theta}}_k\right\|^2\right)$$

$$\leq\frac{d}{\delta_k}\left(2L_2G\sum_{\ell=1}^{m-1}\sum_{j=1}^{\ell-1}\eta_k\lambda^{\tau_k-j}\frac{dG}{\delta_k}+2GM\rho^m+c_8\sum_{j=1}^{m-1}\eta_k\lambda^{\tau_k-j}\frac{dG}{\delta_k}\right)$$

$$+\frac{d}{\delta_k}\frac{L}{2}\left(\sum_{j=1}^{m-1}\eta_k\lambda^{\tau_k-j}\frac{dG}{\delta_k}\right)^2$$

$$<\frac{d}{(1-\lambda)^2}\left(2L_2G^2d+c_8Gd\right)\lambda^{\tau_k-m+1}\frac{\eta_k}{\delta_k^2}+\frac{LG^2d^3}{2(1-\lambda)^2}\lambda^{2(\tau_k-m+1)}\frac{\eta_k^2}{\delta_k^3}+2GMd\frac{\rho^m}{\delta_k}$$

Then it holds that

$$\left\|\sum_{m=1}^{\tau_k}\lambda^{\tau_k-m}\Delta_{k,m}\right\|\leq\sum_{m=1}^{\tau_k}\lambda^{\tau_k-m}\|\Delta_{k,m}\|$$

$$\leq\frac{d}{(1-\lambda)^2}\left(2L_2G^2d+c_8dG\right)\frac{\eta_k}{\delta_k^2}\sum_{m=1}^{\tau_k}\lambda^{2(\tau_k-m)}\lambda$$

$$+\frac{LG^2}{2(1-\lambda)^2}d^3\frac{\eta_k^2}{\delta_k^3}\sum_{m=1}^{\tau_k}\lambda^{3(\tau_k-m)}\lambda^2$$

$$+2GMd\delta_k^{-1}\sum_{m=1}^{\tau_k}\lambda^{\tau_k-m}\rho^m$$

$$\leq(2L_2G^2d+c_8Gd)\frac{d\lambda}{(1-\lambda)^3}\frac{\eta_k}{\delta_k^2}$$

$$+\frac{LG^2}{2}\frac{d^3\lambda^2}{1-\lambda}\frac{\eta_k^2}{\delta_k^3}+2GMd\delta_k^{-1}\tau_k\max\{\rho,\lambda\}^{\tau_k}$$

Finally, provided $\tau_k\geq\log_{\max\{\rho,\lambda\}}(1+k)^{-1}$ and $0<2\alpha-4\beta<1$, we can bound $\mathbf{I}_2(t)$ as follows:

$$\mathbf{I}_2(t)\leq(1-\lambda)\mathbb{E}\sum_{k=0}^{t}\|\nabla\mathcal{L}(\boldsymbol{\theta}_k)\|\cdot\left\|\sum_{m=1}^{\tau_k}\lambda^{\tau_k-m}\Delta_{k,m}\right\|$$

$$\leq(1-\lambda)\mathbb{E}\sum_{k=0}^{t}\|\nabla\mathcal{L}(\boldsymbol{\theta}_k)\|\left[(2L_2G^2d+c_8Gd)\frac{d\lambda}{(1-\lambda)^3}\frac{\eta_k}{\delta_k^2}+\frac{LG^2}{2}\frac{d^3\lambda^2}{1-\lambda}\frac{\eta_k^2}{\delta_k^3}+2GMd\frac{\tau_k}{\delta_k(1+k)}\right]$$

$$\leq\frac{d\lambda}{(1-\lambda)^2}(2L_2G^2d+c_8Gd)\left(\sum_{k=0}^{t}\mathbb{E}\|\nabla\mathcal{L}(\boldsymbol{\theta}_k)\|^2\right)^{1/2}\left(\sum_{k=0}^{t}\frac{\eta_k^2}{\delta_k^4}\right)^{1/2}$$

$$+d^3\lambda^2\frac{LG^2}{2}\left(\sum_{k=0}^{t}\mathbb{E}\|\nabla\mathcal{L}(\boldsymbol{\theta}_k)\|^2\right)^{1/2}\left(\sum_{k=0}^{t}\frac{\eta_k^4}{\delta_k^6}\right)^{1/2}$$

$$+\frac{d}{1-\lambda}GM\left(\sum_{k=0}^{t}\mathbb{E}\|\nabla\mathcal{L}(\boldsymbol{\theta}_k)\|^2\right)^{1/2}\left(\sum_{k=0}^{t}\frac{\tau_k^2}{\delta_k^2(1+k)^2}\right)^{1/2}$$

$$\overset{(b)}{\leq}\frac{d^2\lambda}{(1-\lambda)^2}6(L_2G^2+\sqrt{L}G^{3/2}+L_1G^2)\left(\sum_{k=0}^{t}\mathbb{E}\|\nabla\mathcal{L}(\boldsymbol{\theta}_k)\|^2\right)^{1/2}\left(\sum_{k=0}^{t}\frac{\eta_k^2}{\delta_k^4}\right)^{1/2}$$

$$\leq c_2 \frac{d^2}{(1-\lambda)^2} \left( \frac{1}{1+t} \sum_{k=0}^{t} \mathbb{E} \left\| \nabla \mathcal{L}(\boldsymbol{\theta}_k) \right\|^2 \right)^{1/2} \cdot (1+t)^{1-(\alpha-2\beta)} \quad \forall t \geq \max\{t_1, t_2\}$$

where $c_2 := \frac{\eta_0}{\delta_0^2} \frac{6 \cdot (L_1 G^2 + L_2 G^2 + \sqrt{L} G^{3/2})}{\sqrt{1-2\alpha+4\beta}}$. The inequality (b) holds since $\tau_k = \Theta(\log k)$, $4\alpha - 6\beta > 2\alpha - 4\beta$ and $2 - 2\beta > 2\alpha - 4\beta$, so there exist constants

$$t_1 := \inf_t \left\{ t \geq 0 \,\Big|\, \frac{d^6 \lambda^4 L^2 G^4}{4} \sum_{k=0}^{t} \frac{\eta_k^4}{\delta_k^6} \leq \frac{d^2 \lambda^2 (2L_2 G^2 d + c_8 G d)^2}{(1-\lambda)^4} \sum_{k=0}^{t} \frac{\eta_k^2}{\delta_k^4} \right\} \quad (25)$$

$$t_2 := \inf_t \left\{ t \geq 0 \,\Big|\, d^2 G^2 M^2 \sum_{k=0}^{t} \frac{\tau_k^2}{\delta_k^2 (1+k)^2} \leq \frac{d^2 \lambda^2 (2L_2 G^2 d + c_8 G d)^2}{(1-\lambda)^4} \sum_{k=0}^{t} \frac{\eta_k^2}{\delta_k^4} \right\} \quad (26)$$

In brief, we have

$$\mathbf{I}_2(t) \leq c_2 \frac{d^2}{(1-\lambda)^2} \left( \frac{1}{1+t} \sum_{k=0}^{t} \mathbb{E} \left\| \nabla \mathcal{L}(\boldsymbol{\theta}_k) \right\|^2 \right)^{1/2} \cdot (1+t)^{1-(\alpha-2\beta)} \quad \forall t \geq \max\{t_1, t_2\}$$

$\square$

**Lemma B.3.** *Under Assumption 3.1, 3.2 and* $0 < \beta < 1/2$, *with* $\tau_k \geq \frac{1}{\log 1/\max\{\rho,\lambda\}} \left( \log(1+k) + \log \frac{d}{\delta_0} \right)$, *it holds that*

$$\mathbf{I}_3(t) \leq c_3 \mathcal{A}(t)^{\frac{1}{2}} (1+t)^{1-\beta} \quad (27)$$

*where* $\mathcal{A}(t) := \frac{1}{1+t} \sum_{k=0}^{t} \mathbb{E} \left\| \nabla \mathcal{L}(\boldsymbol{\theta}_k) \right\|^2$ *and constant* $c_3 = \frac{1}{\sqrt{1-2\beta}} \max\{2^{1-\beta} L \delta_0, 2^\beta G \sqrt{1-\beta}\}$.

*Proof.* Recall that $g_k(\boldsymbol{\theta}; u, z) := g_{\delta_k}(\boldsymbol{\theta}; u, z)$ and $\mathcal{L}_k(\boldsymbol{\theta}) := \mathcal{L}_{\delta_k}(\boldsymbol{\theta})$.

$$\mathbf{I}_3(t) = -\sum_{k=0}^{t} \mathbb{E} \left\langle \nabla \mathcal{L}(\boldsymbol{\theta}_k) \,\Big|\, (1-\lambda) \left( \sum_{m=1}^{\tau_k} \lambda^{\tau_k - m} \nabla \mathcal{L}_k(\boldsymbol{\theta}_k) \right) - \nabla \mathcal{L}(\boldsymbol{\theta}_k) \right\rangle$$

$$= -\sum_{k=0}^{t} \mathbb{E} \left\langle \nabla \mathcal{L}(\boldsymbol{\theta}_k) \,\Big|\, \left( (1-\lambda) \sum_{m=1}^{\tau_k} \lambda^{\tau_k - m} \right) \nabla \mathcal{L}_k(\boldsymbol{\theta}_k) - \nabla \mathcal{L}(\boldsymbol{\theta}_k) \right\rangle$$

$$= -\sum_{k=0}^{t} \mathbb{E} \left\langle \nabla \mathcal{L}(\boldsymbol{\theta}_k) \,\big|\, \nabla \mathcal{L}_k(\boldsymbol{\theta}_k) - \nabla \mathcal{L}(\boldsymbol{\theta}_k) \right\rangle - \lambda^{\tau_k} \mathbb{E} \left\langle \nabla \mathcal{L}(\boldsymbol{\theta}_k) \,\big|\, \mathbb{E}_{Z \sim \Pi_{\check{\boldsymbol{\theta}}_k}} [g_k(\boldsymbol{\theta}_k; u_k, Z)] \right\rangle$$

where we apply Lemma E.1 at the last equality.

By triangle inequality, Cauchy-Schwarz inequality and Assumption 3.2, we obtain

$$\mathbf{I}_3(t) \leq \sum_{k=0}^{t} \mathbb{E} \left\| \nabla \mathcal{L}(\boldsymbol{\theta}_k) \right\| \cdot \left\| \nabla \mathcal{L}_k(\boldsymbol{\theta}_k) - \nabla \mathcal{L}(\boldsymbol{\theta}_k) \right\| + \sum_{k=0}^{t} \lambda^{\tau_k} \mathbb{E} \left\| \nabla \mathcal{L}(\boldsymbol{\theta}_k) \right\| \frac{dG}{\delta_k}$$

Provided $\tau_k \geq \frac{\log(1+k) + \log \frac{d}{\delta_0}}{\log 1/\max\{\rho,\lambda\}} \geq \frac{\log \delta_0 / d(1+k)^{-1}}{\log \max\{\rho,\lambda\}} = \log_{\max\{\rho,\lambda\}} \frac{\delta_0}{d} (1+k)^{-1} \geq \log_\lambda \frac{\delta_0}{d} (1+k)^{-1}$, with Lemma E.2 as a consequence of Assumption 3.1, we have

$$\mathbf{I}_3(t) \leq \sum_{k=0}^{t} \mathbb{E} \left\| \nabla \mathcal{L}(\boldsymbol{\theta}_k) \right\| \cdot L \delta_k + \sum_{k=0}^{t} \mathbb{E} \left\| \nabla \mathcal{L}(\boldsymbol{\theta}_k) \right\| \frac{\delta_0}{d} \frac{dG}{\delta_0} (1+k)^{\beta-1}$$

$$= \sum_{k=0}^{t} \mathbb{E} \left\| \nabla \mathcal{L}(\boldsymbol{\theta}_k) \right\| \cdot L \delta_k + G \sum_{k=0}^{t} \mathbb{E} \left\| \nabla \mathcal{L}(\boldsymbol{\theta}_k) \right\| (1+k)^{\beta-1}$$

$$\leq L \left( \sum_{k=0}^{t} \mathbb{E} \left\| \nabla \mathcal{L}(\boldsymbol{\theta}_k) \right\|^2 \right)^{1/2} \left( \sum_{k=0}^{t} \delta_k^2 \right)^{1/2} + G \left( \sum_{k=0}^{t} \mathbb{E} \left\| \nabla \mathcal{L}(\boldsymbol{\theta}_k) \right\|^2 \right)^{1/2} \left( \sum_{k=0}^{t} (1+k)^{2(\beta-1)} \right)^{1/2}$$

444 Since $\beta < 1/2$, it holds that

$$\sum_{k=0}^{t} \delta_k^2 = \sum_{k=0}^{t} \frac{\delta_0^2}{(1+k)^{2\beta}} \leq \frac{\delta_0^2}{1-2\beta} \left[ 1 - 2\beta + (1+t)^{1-2\beta} - 1 \right] \leq \frac{\delta_0^2}{1-2\beta} (1+t)^{1-2\beta}$$

$$\sum_{k=0}^{t} (1+k)^{2(\beta-1)} \leq 1 + \int_0^t (x+1)^{2(\beta-1)} dx < 1 + \frac{1}{1-2\beta}$$

445 Then we can conclude

$$\mathbf{I}_3(t) \leq c_3 \left( \frac{1}{1+t} \sum_{k=0}^{t} \mathbb{E} \left\| \nabla \mathcal{L}(\boldsymbol{\theta}_k) \right\|^2 \right)^{1/2} \cdot (1+t)^{1-\beta}$$

446 where $c_3 := \frac{2}{\sqrt{1-2\beta}} \max\{L\delta_0, G\sqrt{1-\beta}\}$. $\qquad\square$

447 **Lemma B.4.** *Under assumption 3.2 and constraint $0 < \alpha < 1$, it holds that*

$$\mathbf{I}_4(t) \leq c_4 \frac{d^2}{1-\lambda} (1+t)^{1-(\alpha-2\beta)} \tag{28}$$

448 *where constant $c_4 = \frac{\eta_0 L G^2}{\delta_0^2 (2\beta-\alpha+1)}$.*

*Proof.*

$$\begin{aligned}
\mathbf{I}_4(t) &= \frac{(1-\lambda)L}{2} \sum_{k=0}^{t} \eta_k \mathbb{E} \left\| \sum_{m=1}^{\tau_k} \lambda^{\tau_k - m} g_k^{(m)} \right\|^2 \\
&\leq \frac{(1-\lambda)L}{2} \sum_{k=0}^{t} \eta_k \mathbb{E} \left( \sum_{m=1}^{\tau_k} \lambda^{\tau_k - m} \left\| g_k^{(m)} \right\| \right)^2 \\
&\leq \frac{(1-\lambda)L}{2} \sum_{k=0}^{t} \eta_k \left( \sum_{m=1}^{\tau_k} \lambda^{\tau_k - m} \right)^2 \frac{(dG)^2}{\delta_k^2} \\
&\leq \frac{(1-\lambda)L d^2 G^2}{2} \sum_{k=0}^{t} \left( \frac{1 - \lambda^{\tau_k}}{1-\lambda} \right)^2 \frac{\eta_k}{\delta_k^2} \\
&< \frac{d^2 L G^2}{2(1-\lambda)} \sum_{k=0}^{t} \frac{\eta_k}{\delta_k^2}
\end{aligned}$$

449 Recall that $\eta_k = \frac{\eta_0}{(k+1)^\alpha}$, $\delta_k = \frac{\delta_0}{(1+k)^\beta}$ and $\alpha < 1, \beta \geq 0$, it is clear that $\alpha - 2\beta < 1$, so it holds that

$$\sum_{k=0}^{t} \frac{\eta_k}{\delta_k^2} = \frac{\eta_0}{\delta_0^2} \sum_{k=0}^{t} (1+k)^{2\beta-\alpha} \leq \frac{\eta_0}{\delta_0^2} \left( 1 + \int_0^t (1+x)^{2\beta-\alpha} dx \right)$$

$$\leq \frac{\eta_0}{\delta_0^2 (2\beta - \alpha + 1)} \left[ (1+t)^{2\beta-\alpha+1} - \alpha + 2\beta \right] \leq \frac{2\eta_0}{\delta_0^2 (2\beta - \alpha + 1)} (1+t)^{2\beta-\alpha+1}$$

450 In conclusion, we obtain that

$$\mathbf{I}_4(t) \leq d^2 \frac{LG^2}{1-\lambda} \frac{\eta_0}{\delta_0^2 (2\beta - \alpha + 1)} \cdot (1+t)^{2\beta-\alpha+1} = c_4 \frac{d^2}{1-\lambda} (1+t)^{1-(\alpha-2\beta)}$$

451 where $c_4 := \frac{\eta_0}{\delta_0^2} \cdot \frac{LG^2}{2\beta-\alpha+1}$. $\qquad\square$

## C  Proof of Lemma 4.3

*Proof.* Combining Lemmas 4.1 and 4.2, subject to the constraints $0 < \alpha < 1, 0 < \beta \leq 1/2, 0 < 2\alpha - 4\beta \leq 1$, it holds that for any $t \geq \max\{t_1, t_2\}$,

$$
\begin{aligned}
\sum_{k=0}^{t} & \mathbb{E} \|\nabla \mathcal{L}(\boldsymbol{\theta}_k)\|^2 \\
& \leq \mathbf{I}_1(t) + \mathbf{I}_2(t) + \mathbf{I}_3(t) + \mathbf{I}_4(t) \\
& \leq c_1(1-\lambda)(1+t)^{\alpha} + c_2 \frac{d^{5/2}}{(1-\lambda)^2}(1+t)^{1-(\alpha-2\beta)}\mathcal{A}(t)^{1/2} \\
& \quad + c_3(1+t)^{1-\beta}\mathcal{A}(t)^{1/2} + c_4 \frac{d^2}{1-\lambda}(1+t)^{1-(\alpha-2\beta)}
\end{aligned}
$$

Recall $\mathcal{A}(t) := \frac{1}{1+t}\sum_{k=0}^{t}\mathbb{E}\|\nabla\mathcal{L}(\boldsymbol{\theta}_k)\|^2$, above inequality can be rewritten as

$$
\begin{aligned}
\mathcal{A}(t) \leq \frac{1}{1+t}\bigg[ & c_2 \frac{d^{5/2}}{(1-\lambda)^2}(1+t)^{1-(\alpha-2\beta)}\mathcal{A}(t)^{1/2} \\
& + c_3(1+t)^{1-\beta}\mathcal{A}(t)^{1/2} + c_1(1-\lambda)(1+t)^{\alpha} + c_4 \frac{d^2}{1-\lambda}(1+t)^{1-(\alpha-2\beta)}\bigg] \\
= & \left( c_2 \frac{d^{5/2}}{(1-\lambda)^2}(1+t)^{-(\alpha-2\beta)} + c_3(1+t)^{-\beta}\right)\mathcal{A}(t)^{1/2} + c_1(1-\lambda)(1+t)^{-(1-\alpha)} \\
& + c_4 \frac{d^2}{1-\lambda}(1+t)^{-(\alpha-2\beta)}
\end{aligned}
$$

which is a quadratic inequality in $\mathcal{A}(t)^{1/2}$.

Let $x = \mathcal{A}(t)^{1/2}, a = c_2 \frac{d^{5/2}}{(1-\lambda)^2}(1+t)^{-(\alpha-2\beta)} + c_3 t^{-\beta}, b = c_1(1-\lambda)(1+t)^{-(1-\alpha)} + c_4 \frac{d^2}{1-\lambda}(1+t)^{-(\alpha-2\beta)}$, we have $x^2 - ax - b \leq 0$. Since $a, b > 0$, the quadratic has two real roots, denoted as $x_1, x_2$ respectively, and $x_1 < 0 < x_2$. Moreover, we must have $x \leq x_2$, which implies $x \leq \frac{a+\sqrt{a^2+4b}}{2} \leq \frac{a+a+2\sqrt{b}}{2} = a + \sqrt{b}$. Therefore, $\mathcal{A}(t) = x^2 \leq (a+\sqrt{b})^2 \leq 2(a^2 + b)$. Substituting $a, b$ back leads to

$$
\begin{aligned}
\mathcal{A}(t) \leq & 2\left( c_2 \frac{d^{5/2}}{(1-\lambda)^2}(1+t)^{-(\alpha-2\beta)} + c_3(1+t)^{-\beta}\right)^2 + 2c_1(1-\lambda)(1+t)^{-(1-\alpha)} \\
& + 2c_4 \frac{d^2}{1-\lambda}(1+t)^{-(\alpha-2\beta)} \\
\overset{(a)}{\leq} & 4c_2^2 \frac{d^5}{(1-\lambda)^4}(1+t)^{-2(\alpha-2\beta)} + 4c_3^2(1+t)^{-2\beta} + 2c_1(1-\lambda)(1+t)^{-(1-\alpha)} \\
& + 2c_4 \frac{d^2}{1-\lambda}(1+t)^{-(\alpha-2\beta)} \\
\leq & 4c_3^2(1+t)^{-2\beta} + 2c_1(1-\lambda)(1+t)^{-(1-\alpha)} + 4c_4 \frac{d^2}{1-\lambda}(1+t)^{-(\alpha-2\beta)},
\end{aligned}
$$

where inequality (a) is due to the fact $(x+y)^2 \leq 2(x^2+y^2)$, the last inequality holds because there exists sufficiently large constant $t_3$ such that, $4c_2^2 \frac{d^5}{(1-\lambda)^4}(1+t)^{-2(\alpha-2\beta)} \leq 2c_4 \frac{d^2}{1-\lambda}(1+t)^{-(\alpha-2\beta)} \forall t \geq t_3$ given $\alpha > 2\beta$. Therefore, set $t_0 := \max\{t_1, t_2, t_3\}$, then for all $t \geq t_0$, we have

$$
\begin{aligned}
\mathcal{A}(t) \leq & 4\max\{c_1(1-\lambda), c_3^2, c_4 \frac{d^2}{1-\lambda}\} \cdot \left((1+t)^{-2\beta} + (1+t)^{-(1-\alpha)} + (1+t)^{-(\alpha-2\beta)}\right) \\
\leq & 12\max\{c_1(1-\lambda), c_3^2, c_4 \frac{d^2}{1-\lambda}\}(1+t)^{-\min\{2\beta, 1-\alpha, \alpha-2\beta\}}
\end{aligned}
$$

Recall that constant $c_1$ contains $1/\eta_0$, $c_3$ contains $\delta_0$, $c_4$ contains $\eta_0/\delta_0^2$, , thus we can set $\delta_0 = d^{1/3}, \eta_0 = d^{-2/3}$, which yields

$$
\mathcal{A}(t) \leq 12\max\{c_5(1-\lambda), c_6, \frac{c_7}{1-\lambda}\}d^{2/3}(1+t)^{-\min\{2\beta, 1-\alpha, \alpha-2\beta\}}
$$

where constants

$$c_5 = 2G, \quad c_6 = \frac{4 \max\{L^2, G^2(1-\beta)\}}{1-2\beta}, \quad c_7 = \frac{LG^2}{2\beta - \alpha + 1}$$

do not contain $\eta_0$ and $\delta_0$. Moreover, note that $\max_{\alpha,\beta} \min\{2\beta, 1-\alpha, \alpha - 2\beta\} = \frac{1}{3}$, thus it holds

$$\frac{1}{1+T} \sum_{k=0}^{T} \mathbb{E} \|\nabla \mathcal{L}(\boldsymbol{\theta})_k\|^2 \leq 12 \max\{c_5(1-\lambda), c_6, \frac{c_7}{1-\lambda}\} d^{2/3}(1+T)^{-1/3}$$

where the rate $\mathcal{O}(1/T^{1/3})$ can be attained by choosing $\alpha = \frac{2}{3}$, $\beta = \frac{1}{6}$. This immediately leads to Theorem 3.1 by observing

$$\min_{0 \leq k \leq T} \mathbb{E} \|\nabla \mathcal{L}(\boldsymbol{\theta}_k)\|^2 \leq \frac{1}{1+T} \sum_{k=0}^{T} \mathbb{E} \|\nabla \mathcal{L}(\boldsymbol{\theta}_k)\|^2 .$$

$\square$

# D   Non-smooth Optimization Analysis

In this section, we aim to apply our algorithm to non-smooth performative risk optimization problem and analyze its convergence behavior. Before presenting the theorem, we need the following Lipschitz loss assumption D.1.

**Assumption D.1.** **(Lipschitz Loss)** There exists constant $L_0 > 0$ such that

$$|\ell(\boldsymbol{\theta}_1; z) - \ell(\boldsymbol{\theta}_2; z)| \leq L_0 \|\boldsymbol{\theta}_1 - \boldsymbol{\theta}_2\|, \ \forall \boldsymbol{\theta}_1, \boldsymbol{\theta}_2 \in \mathbb{R}^d, \ \forall z \in \mathsf{Z}$$

Under Lipschitz loss assumption D.1 and some regularity condition, one can show that the performative risk is also Lipschitz continuous, which is stated as follows.

**Lemma D.1.** *Under Assumption D.1, 3.2, 3.3, the performative risk $\mathcal{L}(\boldsymbol{\theta})$ is $(L_0 + 2L_1 G)$-Lipschitz continuous.*

Under non-smooth settings, the convergence behavior can be characterized in both squared gradient norm and proximity gap. Now, we are ready to show the following theorem:

**Theorem D.1.** (DFO $(\lambda)$ **for Non-smooth Optimization**) *Under Assumption D.1, 3.2, 3.3, 3.4, 3.5, with two time-scale step sizes $\eta_k = \eta_0(1+k)^{-\alpha}, \delta_k = d(1+k)^{-\beta}, \tau_k \geq \frac{\log(1+k)}{\log 1/\max\{\rho, \lambda\}}$, where $\alpha, \beta$ satisfies $0 < 3\beta < \alpha < 1$, there exists a constant $t_4$ such that, the iterates $\{\boldsymbol{\theta}_k\}_{k \geq 1}$ satisfies for all $T \geq t_4$*

$$\frac{1}{1+T} \sum_{k=0}^{T} \mathbb{E} \|\nabla \mathcal{L}_{\delta_k}(\boldsymbol{\theta}_k)\|^2 = \mathcal{O}(T^{-\min\{1-\alpha, \alpha-3\beta\}})$$

*and the following error estimate holds for all $T > 0$ and $\boldsymbol{\theta} \in \mathbb{R}^d$*

$$\frac{1}{1+T} \sum_{k=0}^{T} \mathbb{E}|\mathcal{L}_{\delta_k}(\boldsymbol{\theta}) - \mathcal{L}(\boldsymbol{\theta})| = \mathcal{O}(T^{-\beta})$$

**Corollary D.1.** *($\epsilon$-stationarity, $\mu$-proximity) Suppose Assumptions of Theorem D.1 hold. Fix any $\epsilon, \mu > 0$, the estimate $\frac{1}{1+T} \sum_{k=0}^{T} \mathbb{E} \|\nabla \mathcal{L}_{\delta_k}(\boldsymbol{\theta}_k)\|^2 \leq \epsilon$ and $\frac{1}{1+T} \sum_{k=0}^{T} \mathbb{E} |\mathcal{L}_{\delta_k}(\boldsymbol{\theta}_k) - \mathcal{L}(\boldsymbol{\theta}_k)| \leq \mu$ holds for all*

$$T \geq \max\{\mathcal{O}(1/\epsilon^4), \mathcal{O}(1/\mu^6)\}$$

Next, we present the proof of Theorem D.1.

*Proof.* This proof resembles the proof of Lemma 4.2, where we reinterpret $\sum_{k=0}^{t} \mathbb{E} \|\nabla \mathcal{L}(\boldsymbol{\theta}_k)\|^2$ as $\sum_{k=0}^{t} \mathbb{E} \|\nabla \mathcal{L}_{\delta_k}(\boldsymbol{\theta}_k)\|^2$, and $\mathcal{L}(\boldsymbol{\theta}_k)$ as $\mathcal{L}_{\delta_k}(\boldsymbol{\theta}_k)$, with additional bias terms that, as we shall prove, are not dominant.

Due to Lemma D.1, $\mathcal{L}(\boldsymbol{\theta})$ is $(L_0 + 2L_1 G)$-Lipschitz. Then by Lemma E.1, $\mathcal{L}_\delta(\boldsymbol{\theta})$ is $\frac{d}{\delta}(L_0 + 2L_1 G)$-smooth for all $\delta > 0$. Similar to Lemma 4.1, we have

$$\mathcal{L}_{\delta_k}(\boldsymbol{\theta}_{k+1}) - \mathcal{L}_{\delta_k}(\boldsymbol{\theta}_k) + \frac{\eta_k}{1-\lambda} \left\langle \nabla \mathcal{L}_{\delta_k}(\boldsymbol{\theta}_k) \,|\, (1-\lambda) \sum_{m=1}^{\tau_k} \lambda^{\tau_k - m} g_k^{(m)} \right\rangle$$

$$\leq \frac{d(L_0 + 2L_1 G)}{2\delta_k} \eta_k^2 \left\| \sum_{m=1}^{\tau_k} \lambda^{\tau_k - m} g_k^{(m)} \right\|^2$$

By adding, subtracting and rearranging terms, after taking conditional expectation on $\mathcal{F}^{k-1}$, it holds that

$$\frac{\eta_k}{1-\lambda} \left\| \nabla \mathcal{L}_{\delta_k}(\boldsymbol{\theta}_k) \right\|^2 \leq \mathbb{E}_{\mathcal{F}^{k-1}} \left[ \mathcal{L}_{\delta_k}(\boldsymbol{\theta}_k) - \mathcal{L}_{\delta_{k+1}}(\boldsymbol{\theta}_{k+1}) + \mathcal{L}_{\delta_{k+1}}(\boldsymbol{\theta}_{k+1}) - \mathcal{L}_{\delta_k}(\boldsymbol{\theta}_{k+1}) \right]$$

$$+ \frac{\eta_k}{1-\lambda} \mathbb{E}_{\mathcal{F}^{k-1}} \left\langle \nabla \mathcal{L}_{\delta_k}(\boldsymbol{\theta}_k) \,|\, \nabla \mathcal{L}_{\delta_k}(\boldsymbol{\theta}_k) - (1-\lambda) \sum_{m=1}^{\tau_k} \lambda^{\tau_k - m} g_k^{(m)} \right\rangle$$

$$+ \frac{d}{2\delta_k}(L_0 + 2L_1 G)\eta_k^2 \mathbb{E}_{\mathcal{F}^{k-1}} \left\| \sum_{m=1}^{\tau_k} \lambda^{\tau_k - m} g_k^{(m)} \right\|^2$$

By Lemma E.1, we have $\mathbb{E}_{Z \sim \Pi_{\hat{\boldsymbol{\theta}}_k}, u_k}[g_{\delta_k}(\boldsymbol{\theta}_k; u_k, Z)] = \nabla \mathcal{L}_{\delta_k}(\boldsymbol{\theta}_k)$, then by dividing and summing over $k$, it holds that

$$(1-\lambda) \sum_{k=0}^{t} \mathbb{E} \left\| \nabla \mathcal{L}_{\delta_k}(\boldsymbol{\theta}_k) \right\|^2$$

$$\leq \sum_{k=0}^{t} \frac{1-\lambda}{\eta_k} \mathbb{E} \left[ \mathcal{L}_{\delta_k}(\boldsymbol{\theta}_k) - \mathcal{L}_{\delta_{k+1}}(\boldsymbol{\theta}_{k+1}) + \mathcal{L}_{\delta_{k+1}}(\boldsymbol{\theta}_{k+1}) - \mathcal{L}_{\delta_k}(\boldsymbol{\theta}_{k+1}) \right]$$

$$+ (1-\lambda) \sum_{k=0}^{t} \mathbb{E} \left\langle \nabla \mathcal{L}_{\delta_k}(\boldsymbol{\theta}_k) \,|\, \sum_{m=1}^{\tau_k} \lambda^{\tau_k - m} \left( \mathbb{E}_{Z \sim \Pi_{\hat{\boldsymbol{\theta}}_k}}[g_{\delta_k}(\boldsymbol{\theta}_k; u_k, Z)] - g_k^{(m)} \right) \right\rangle$$

$$+ \sum_{k=0}^{t} \lambda^{\tau_k} \mathbb{E} \left\| \nabla \mathcal{L}_{\delta_k}(\boldsymbol{\theta}_k) \right\|^2$$

$$+ \frac{d(L_0 + 2L_1 G)(1-\lambda)}{2} \sum_{k=0}^{t} \frac{\eta_k}{\delta_k} \mathbb{E} \left\| \sum_{m=1}^{\tau_k} \lambda^{\tau_k - m} g_k^{(m)} \right\|^2$$

$$:= \mathbf{I}_5(t) + \mathbf{I}_6(t) + \mathbf{I}_7(t) + \mathbf{I}_8(t)$$

After splitting RHS into $\mathbf{I}_5(t), \mathbf{I}_6(t), \mathbf{I}_7(t), \mathbf{I}_8(t)$, we can bound them separately.

Under Assumption 3.2 and the estimate $\delta_k - \delta_{k+1} = \Theta(k^{-\beta-1})$, it holds that

$$\mathbf{I}_5(t) = (1-\lambda) \sum_{k=0}^{t} \frac{1}{\eta_k} \mathbb{E} \left[ \mathcal{L}_{\delta_k}(\boldsymbol{\theta}_k) - \mathcal{L}_{\delta_{k+1}}(\boldsymbol{\theta}_{k+1}) \right] + (1-\lambda) \sum_{k=0}^{t} \frac{1}{\eta_k} \mathbb{E} \left[ \mathcal{L}_{\delta_{k+1}}(\boldsymbol{\theta}_{k+1}) - \mathcal{L}_{\delta_k}(\boldsymbol{\theta}_{k+1}) \right]$$

$$\overset{(a)}{\leq} (1-\lambda)G \frac{2}{\eta_{t+1}} + (1-\lambda) \sum_{k=0}^{t} \mathbb{E} \frac{\mathcal{L}_{\delta_{k+1}}(\boldsymbol{\theta}_k) - \mathcal{L}_{\delta_k}(\boldsymbol{\theta}_k)}{\eta_k}$$

$$\overset{(b)}{\leq} (1-\lambda)G \frac{2}{\eta_{t+1}} + (1-\lambda)(L_0 + 2L_1 G) \sum_{k=0}^{t} \frac{\delta_k - \delta_{k+1}}{\eta_k}$$

$$= \mathcal{O} \left( (1+t)^\alpha + (1+t)^{\alpha-\beta} \right) = \mathcal{O} \left( (1+t)^\alpha \right)$$

where we apply the summation by part in inequality (a) as in Lemma B.1, and use the fact $|\mathcal{L}_{\delta_1}(\boldsymbol{\theta}) - \mathcal{L}_{\delta_2}(\boldsymbol{\theta})| \leq \mathbb{E}_w |\mathcal{L}(\boldsymbol{\theta} + \delta_1 w) - \mathcal{L}(\boldsymbol{\theta} + \delta_2 w)| \leq (L_0 + 2L_1 G)|\delta_1 - \delta_2|$ in inequality (e), as a consequence of Lipschitz continuity.

506 As for $\mathbf{I}_6(t)$, if we let $\mathcal{B}(t) := \frac{1}{1+t}\sum_{k=0}^t \mathbb{E}\left\|\nabla\mathcal{L}_{\delta_k}(\boldsymbol{\theta}_k)\right\|^2$, by definition of $g_k^{(m)}$, we can split the
507 term as follows

$$\mathbb{E}_{\mathcal{F}^{k-1}}\frac{d}{\delta_k}\left(\mathbb{E}_{Z\sim\Pi_{\check{\boldsymbol{\theta}}_k}}[\ell(\check{\boldsymbol{\theta}}_k;Z)|\check{\boldsymbol{\theta}}_k] - \mathbb{E}[\ell(\check{\boldsymbol{\theta}}_k^{(m)};Z_k^{(m)})|\check{\boldsymbol{\theta}}_k^{(m)},Z_k^{(0)}]\right)$$

$$= \mathbb{E}_{\mathcal{F}^{k-1}}\frac{d}{\delta_k}\mathbb{E}_{Z\sim\Pi_{\check{\boldsymbol{\theta}}_k}}\left[\ell(\check{\boldsymbol{\theta}}_k;Z) - \ell(\check{\boldsymbol{\theta}}_k^{(m)};Z)|\check{\boldsymbol{\theta}}_k^{(m)},\check{\boldsymbol{\theta}}_k\right]$$

$$+ \mathbb{E}_{\mathcal{F}^{k-1}}\frac{d}{\delta_k}\left(\mathbb{E}_{Z\sim\Pi_{\check{\boldsymbol{\theta}}_k}}[\ell(\check{\boldsymbol{\theta}}_k^{(m)};Z)|\check{\boldsymbol{\theta}}_k^{(m)}] - \mathbb{E}_{\tilde{Z}_k^{(m)}}[\ell(\check{\boldsymbol{\theta}}_k^{(m)};\tilde{Z}_k^{(m)})|\check{\boldsymbol{\theta}}_k^{(m)},\tilde{Z}_k^{(0)}]\right)$$

$$+ \mathbb{E}_{\mathcal{F}^{k-1}}\frac{d}{\delta_k}\left(\mathbb{E}_{\tilde{Z}_k^{(m)}}[\ell(\check{\boldsymbol{\theta}}_k^{(m)};\tilde{Z}_k^{(m)})|\check{\boldsymbol{\theta}}_k^{(m)},\tilde{Z}_k^{(0)}] - \mathbb{E}[\ell(\check{\boldsymbol{\theta}}_k^{(m)};Z_k^{(m)})|\check{\boldsymbol{\theta}}_k^{(m)},Z_k^{(0)}]\right)$$

508 By applying Jensen's inequality and triangle inequality according to the above splitting, it holds that

$$\left\|\mathbb{E}_{\mathcal{F}^{k-1}}\mathbb{E}_{Z\sim\Pi_{\check{\boldsymbol{\theta}}_k}}[g_{\delta_k}(\boldsymbol{\theta}_k;u_k,Z)] - g_k^{(m)}\right\|$$

$$= \left|\frac{d}{\delta_k}|\mathbb{E}_{\mathcal{F}^{k-1}}\mathbb{E}_{Z\sim\Pi_{\check{\boldsymbol{\theta}}_k}}[\ell(\check{\boldsymbol{\theta}}_k;Z)|\check{\boldsymbol{\theta}}_k] - \mathbb{E}[\ell(\check{\boldsymbol{\theta}}_k^{(m)};Z_k^{(m)})|\check{\boldsymbol{\theta}}_k^{(m)},Z_k^{(0)}]\right|$$

$$\leq \mathbb{E}_{\mathcal{F}^{k-1}}\frac{d}{\delta_k}\left|\mathbb{E}_{Z\sim\Pi_{\check{\boldsymbol{\theta}}_k}}[\ell(\check{\boldsymbol{\theta}}_k;Z)|\check{\boldsymbol{\theta}}_k] - \mathbb{E}[\ell(\check{\boldsymbol{\theta}}_k^{(m)};Z_k^{(m)})|\check{\boldsymbol{\theta}}_k^{(m)},Z_k^{(0)}]\right|$$

$$\leq \mathbb{E}_{\mathcal{F}^{k-1}}\frac{d}{\delta_k}\left|\mathbb{E}_{Z\sim\Pi_{\check{\boldsymbol{\theta}}_k}}\left[\ell(\check{\boldsymbol{\theta}}_k;Z) - \ell(\check{\boldsymbol{\theta}}_k^{(m)};Z)|\check{\boldsymbol{\theta}}_k^{(m)},\check{\boldsymbol{\theta}}_k\right]\right|$$

$$+ \mathbb{E}_{\mathcal{F}^{k-1}}\frac{d}{\delta_k}\left|\mathbb{E}_{Z\sim\Pi_{\check{\boldsymbol{\theta}}_k}}[\ell(\check{\boldsymbol{\theta}}_k^{(m)};Z)|\check{\boldsymbol{\theta}}_k^{(m)}] - \mathbb{E}_{\tilde{Z}_k^{(m)}}[\ell(\check{\boldsymbol{\theta}}_k^{(m)};\tilde{Z}_k^{(m)})|\check{\boldsymbol{\theta}}_k^{(m)},\tilde{Z}_k^{(0)}]\right|$$

$$+ \mathbb{E}_{\mathcal{F}^{k-1}}\frac{d}{\delta_k}\left|\mathbb{E}_{\tilde{Z}_k^{(m)}}[\ell(\check{\boldsymbol{\theta}}_k^{(m)};\tilde{Z}_k^{(m)})|\check{\boldsymbol{\theta}}_k^{(m)},\tilde{Z}_k^{(0)}] - \mathbb{E}[\ell(\check{\boldsymbol{\theta}}_k^{(m)};Z_k^{(m)})|\check{\boldsymbol{\theta}}_k^{(m)},Z_k^{(0)}]\right|$$

$$\overset{(c)}{\leq} \frac{d}{\delta_k}\mathbb{E}_{\mathcal{F}^{k-1}}L_0\left\|\check{\boldsymbol{\theta}}_k^{(m)} - \check{\boldsymbol{\theta}}_k\right\|$$

$$+ \frac{2dG}{\delta_k}\mathbb{E}_{\mathcal{F}^{k-1}}\boldsymbol{\delta}_{\mathrm{TV}}\left(\Pi_{\boldsymbol{\theta}_k},\mathbb{P}(\hat{Z}_k^{(m)}\in\cdot|\check{\boldsymbol{\theta}}_k^{(0)},\hat{Z}_k^{(0)})\right)$$

$$+ \frac{2dG}{\delta_k}\mathbb{E}_{\mathcal{F}^{k-1}}\boldsymbol{\delta}_{\mathrm{TV}}\left(\mathbb{P}(\hat{Z}_k^{(m)}\in\cdot|\check{\boldsymbol{\theta}}_k^{(0)},\hat{Z}_k^{(0)}),\mathbb{P}(Z_k^{(m)}\in\cdot|\check{\boldsymbol{\theta}}_k^{(0)},Z_k^{(0)})\right)$$

$$\overset{(d)}{\leq} \frac{dL_0}{\delta_k}\mathbb{E}_{\mathcal{F}^{k-1}}\left\|\check{\boldsymbol{\theta}}_k^{(m)} - \check{\boldsymbol{\theta}}_k\right\| + \frac{2dG}{\delta_k}M\rho^m + \frac{2dL_2G}{\delta_k}\mathbb{E}_{\mathcal{F}^{k-1}}\sum_{\ell=1}^{m-1}\left\|\check{\boldsymbol{\theta}}_k^{(\ell)} - \check{\boldsymbol{\theta}}_k\right\|$$

$$\leq \frac{dL_0}{\delta_k}dG\sum_{j=1}^{m-1}\lambda^{\tau_k-j}\frac{\eta_k}{\delta_k} + \frac{2dGM}{\delta_k}\rho^m + \frac{2dL_2G}{\delta_k}dG\sum_{\ell=1}^{m-1}\sum_{j=1}^{\ell-1}\lambda^{\tau_k-j}\frac{\eta_k}{\delta_k}$$

$$< d^2L_0G\frac{\eta_k}{\delta_k^2}\frac{\lambda^{\tau_k-m+1}}{1-\lambda} + \frac{2dGM}{\delta_k}\rho^m + 2d^2L_2G^2\frac{\eta_k}{\delta_k^2}\frac{\lambda^{\tau_k-m+2}}{(1-\lambda)^2}$$

509 where inequality (c) is due to Lipschitzness of decoupled risk, inequality (d) is due to Assumption 3.4
510 and Lemma E.4 (a consequence of Assumption 3.5). Given $\tau_k \geq \frac{\log(1+k)}{\log 1/\max\{\rho,\lambda\}}$, then the following
511 deterministic bound holds for all $k > 0$,

$$\mathbb{E}_{\mathcal{F}^{k-1}}\sum_{m=1}^{\tau_k}\lambda^{\tau_k-m}\left\|\mathbb{E}_{Z\sim\Pi_{\check{\boldsymbol{\theta}}_k}}[g_{\delta_k}(\boldsymbol{\theta}_k;u_k,Z)] - g_k^{(m)}\right\|$$

$$\leq d^2L_0G\frac{\eta_k}{\delta_k^2}\frac{\lambda}{1-\lambda}\sum_{m=1}^{\tau_k}\lambda^{\tau_k-m} + 2d^2L_2G^2\frac{\eta_k}{\delta_k^2}\frac{\lambda^2}{1-\lambda}\sum_{m=1}^{\tau_k}\lambda^{\tau_k-m}$$

$$+ 2dGM/\delta_k\sum_{m=1}^{\tau_k}\rho^m\lambda^{\tau_k-m}$$

$$< d^2\frac{\lambda}{(1-\lambda)^2}L_0G\frac{\eta_k}{\delta_k^2} + 2d^2\frac{\lambda^2}{(1-\lambda)^2}L_2G^2\frac{\eta_k}{\delta_k^2} + 2dGM/\delta_k\sum_{m=1}^{\tau_k}\max\{\rho,\lambda\}^{\tau_k}$$

$$\leq d^2 \frac{\lambda}{(1-\lambda)^2} L_0 G \frac{\eta_k}{\delta_k^2} + 2d^2 \frac{\lambda^2}{(1-\lambda)^2} L_2 G^2 \frac{\eta_k}{\delta_k^2} + 2dGM \frac{\tau_k}{(1+k)\delta_k}$$

512  So for sufficiently large $t$, it holds that

$$\mathbf{I}_6(t) \leq (1-\lambda) \sum_{k=0}^{t} \mathbb{E} \left\| \nabla \mathcal{L}(\boldsymbol{\theta}_k) \right\| \mathbb{E}_{\mathcal{F}^{k-1}} \left\| \sum_{m=1}^{\tau_k} \lambda^{\tau_k - m} \mathbb{E}_{Z \sim \Pi_{\hat{\boldsymbol{\theta}}_k}} [g_{\delta_k}(\boldsymbol{\theta}_k; u_k, Z)] - g_k^{(m)} \right\|$$

$$\leq (1-\lambda) \sum_{k=0}^{t} \mathbb{E} \left\| \nabla \mathcal{L}(\boldsymbol{\theta}_k) \right\| \sum_{m=1}^{\tau_k} \lambda^{\tau_k - m} \mathbb{E}_{\mathcal{F}^{k-1}} \left\| \mathbb{E}_{Z \sim \Pi_{\hat{\boldsymbol{\theta}}_k}} [g_{\delta_k}(\boldsymbol{\theta}_k; u_k, Z)] - g_k^{(m)} \right\|$$

$$\leq \sum_{k=0}^{t} \mathbb{E} \left\| \nabla \mathcal{L}(\boldsymbol{\theta}_k) \right\| d^2 \frac{\lambda}{1-\lambda} ((L_0 + 2L_1 G)G + 2L_1 G + 2\lambda L_2 G^2) \frac{\eta_k}{\delta_k^2}$$

$$+ \sum_{k=0}^{t} \mathbb{E} \left\| \nabla \mathcal{L}(\boldsymbol{\theta}_k) \right\| 2dGM \frac{\tau_k}{(1+k)\delta_k}$$

$$\leq \sum_{k=0}^{t} \mathbb{E} \left\| \nabla \mathcal{L}(\boldsymbol{\theta}_k) \right\| d^2 \frac{2\lambda}{(1-\lambda)^2} (L_0 G + 2L_1 G + 2\lambda L_2 G^2) \frac{\eta_k}{\delta_k^2}$$

$$= d^2 \frac{2\lambda}{(1-\lambda)^2} (L_0 G + 2L_1 G + 2\lambda L_2 G^2) \sum_{k=0}^{t} \mathbb{E} \left\| \nabla \mathcal{L}(\boldsymbol{\theta}_k) \right\| \frac{\eta_k}{\delta_k^2}$$

$$\leq d^2 \frac{2\lambda}{(1-\lambda)^2} (L_0 G + 2L_1 G + 2\lambda L_2 G^2) \left( \sum_{k=0}^{t} \mathbb{E} \left\| \nabla \mathcal{L}(\boldsymbol{\theta}_k) \right\|^2 \right)^{1/2} \left( \sum_{k=0}^{t} \frac{\eta_k^2}{\delta_k^4} \right)^{1/2}$$

$$\leq c_9 d^2 \mathcal{B}(t)^{1/2} (1+t)^{\frac{1}{2}+\frac{1}{2}-(\alpha-2\beta)}$$

513  Therefore, there exists a constant $c_9 > 0$ such that

$$\mathbf{I}_6(t) \leq c_9 d^2 \mathcal{B}(t)^{1/2} (1+t)^{1-(\alpha-2\beta)}$$

514  where there is an extra $\beta/2$ in exponent because the $L$ in $c_2$ is now a variable $d(L_0 + 2L_1 G)/\delta_k$.

515  For $(L_0 + 2L_1 G)$-Lipschitz continuous $\mathcal{L}(\boldsymbol{\theta})$, for all $\delta > 0$ it holds that $\|\nabla \mathcal{L}_\delta(\boldsymbol{\theta})\| \leq (L_0 + 2L_1 G)$.

516  Given $\tau_k \geq \frac{\log(1+k)}{\log 1/\max\{\rho,\lambda\}}$, it holds that $\lambda^{\tau_k} \mathbb{E} \|\nabla \mathcal{L}_{\delta_k}(\boldsymbol{\theta}_k)\|^2 \leq \frac{dL^2}{\delta_0(1+k)}$, then $\mathbf{I}_7(t)$ can be bounded

517  as follows

$$\mathbf{I}_7(t) \leq \frac{dL^2}{\delta_0} \sum_{k=0}^{t} (1+k)^{-1} = \mathcal{O}(\log(1+t))$$

518  $\mathbf{I}_8(t)$ is similar to $\mathbf{I}_4(t)$. For all $0 \leq k \leq t, 1 \leq m \leq \tau_k$, it holds that $\left\| g_k^{(m)} \right\| \leq \frac{dG}{\delta_k}$, which implies

$$\mathbf{I}_8(t) \leq (1-\lambda) \frac{d(L_0 + 2L_1 G)}{2} \sum_{k=0}^{t} \frac{\eta_k}{\delta_k} \mathbb{E} \left\| \sum_{m=1}^{\tau_k} \lambda^{\tau_k - m} g_k^{(m)} \right\|^2$$

$$\leq (1-\lambda) \frac{d(L_0 + 2L_1 G)}{2} \sum_{k=0}^{t} \frac{\eta_k}{\delta_k} \mathbb{E} \left( \sum_{m=1}^{\tau_k} \lambda^{\tau_k - m} \left\| g_k^{(m)} \right\| \right)^2$$

$$\leq (1-\lambda) \frac{d(L_0 + 2L_1 G)}{2} \sum_{k=0}^{t} \frac{\eta_k}{\delta_k} \left( \sum_{m=1}^{\tau_k} \lambda^{\tau_k - m} \frac{dG}{\delta_k} \right)^2$$

$$= (1-\lambda) \frac{d^3 (L_0 + 2L_1 G) G^2}{2} \sum_{k=0}^{t} \frac{\eta_k}{\delta_k^3} \left( \sum_{m=1}^{\tau_k} \lambda^{\tau_k - m} \right)^2$$

$$\leq \frac{d^3 (L_0 + 2L_1 G) G^2}{2(1-\lambda)} \sum_{k=0}^{t} \frac{\eta_k}{\delta_k^3} \leq c_{10} (1+t)^{1-(\alpha-3\beta)}$$

519  where $c_{10} > 0$ is a constant hiding the factor $\frac{\eta_0}{\delta_0^3}$.

520 Applying quadratic technique in Lemma 4.3, and for all $\alpha, \beta$ satisfying $0 < 3\beta < \alpha < 1$, it is clear
521 that only $\mathbf{I}_5(t)$ and $\mathbf{I}_8(t)$ contribute to the asymptotic rate, so for all $t \geq t_4$ (for some constant $t_4 > 0$),
522 we have

$$\frac{1}{1+T} \sum_{k=0}^{T} \mathbb{E} \left\| \nabla \mathcal{L}_{\delta_k}(\boldsymbol{\theta}_k) \right\|^2 = \mathcal{O}(T^{-\min\{1-\alpha, \alpha-3\beta\}})$$

523 The error estimate directly follows from Lemma D.1. $\qquad\square$

## E Auxiliary Lemmas

525 **Lemma E.1. (Smoothing)** *For continuous $\mathcal{L}(\boldsymbol{\theta}) : \mathbb{R}^d \to \mathbb{R}$, its smoothed approximation $\mathcal{L}_\delta(\boldsymbol{\theta}) :=$*
526 $\mathbb{E}_{w \sim Unif(\mathbb{B}^d)}[\mathcal{L}(\boldsymbol{\theta} + \delta w)]$ *is differentiable, and it holds that*

$$\mathbb{E}_{\substack{u \sim Unif(\mathbb{S}^{d-1}), \\ Z \sim \Pi_{\boldsymbol{\theta} + \delta u}}}[g_\delta(\boldsymbol{\theta}; u, Z)] = \nabla \mathcal{L}_\delta(\boldsymbol{\theta})$$

527 *Moreover, if $\mathcal{L}(\boldsymbol{\theta})$ is $\bar{L}$-Lipschitz continuous, then $\mathcal{L}_\delta(\boldsymbol{\theta})$ is $\frac{d}{\delta}\bar{L}$-smooth.*

528 *Proof.* The first fact follows from (generalized) Stoke's theorem. Given continuous $\mathcal{L}(\boldsymbol{\theta})$, it holds
529 that

$$\nabla \int_{\delta \mathbb{B}^d} \mathcal{L}(\boldsymbol{\theta} + v) dv = \int_{\delta \mathbb{S}^{d-1}} \mathcal{L}(\boldsymbol{\theta} + r) \frac{r}{\|r\|} dr \tag{29}$$

530 Observe that the RHS of Equation (29) is continuous in $\boldsymbol{\theta}$, which implies $\mathcal{L}_\delta(\boldsymbol{\theta}) = \frac{1}{\mathrm{vol}(\delta \mathbb{B}^d)} \int_{\delta \mathbb{B}^d} \mathcal{L}(\boldsymbol{\theta} +$
531 $v) dv$ is differentiable. Note that the volume to surface area ratio of $\delta \mathbb{B}^d$ is $\delta/d$, so it follows from
532 Equation (29) that

$$\nabla \mathcal{L}_\delta(\boldsymbol{\theta}) = \frac{\mathrm{vol}(\delta \mathbb{S}^{d-1})}{\mathrm{vol}(\delta \mathbb{B}^d)} \int_{\delta \mathbb{S}^{d-1}} \mathcal{L}(\boldsymbol{\theta} + r) \frac{r}{\mathrm{vol}(\delta \mathbb{S}^{d-1}) \|r\|} dr = \frac{d}{\delta} \mathbb{E}_{u \sim \mathrm{Unif}(\mathbb{S}^{d-1})}[\mathcal{L}(\boldsymbol{\theta} + \delta u) u]$$

$$= \mathbb{E}_{u \sim \mathrm{Unif}(\mathbb{S}^{d-1})} \mathbb{E}_{Z \sim \pi_{\boldsymbol{\theta} + \delta u}}[\frac{d}{\delta} \ell(\boldsymbol{\theta} + \delta u; Z) u] = \mathbb{E}_{\substack{u \sim \mathrm{Unif}(\mathbb{S}^{d-1}), \\ Z \sim \Pi_{\boldsymbol{\theta} + \delta u}}}[g_\delta(\boldsymbol{\theta}; u, Z)]$$

533 where we use the definition of $g_\delta(\boldsymbol{\theta}; u, z)$ in the last equality.

534 If further assuming $\mathcal{L}(\boldsymbol{\theta})$ is $\bar{L}$-Lipschitz continuous, then we obtain

$$\|\nabla \mathcal{L}_\delta(\boldsymbol{\theta}_1) - \nabla \mathcal{L}_\delta(\boldsymbol{\theta}_2)\| = \frac{d}{\delta} \cdot \left\| \frac{1}{\mathrm{vol}(\mathbb{S}^{d-1})} \int_{\mathbb{S}^{d-1}} [\mathcal{L}(\boldsymbol{\theta}_1 + \delta u) - \mathcal{L}(\boldsymbol{\theta}_2 + \delta u)] u du \right\|$$

$$\leq \frac{d}{\delta} \cdot \bar{L} \|\boldsymbol{\theta}_1 - \boldsymbol{\theta}_2\|.$$

535 $\qquad\square$

536 **Lemma E.2. ($\mathcal{O}(\delta)$-Biased Gradient Estimation)**

537 *Under Assumptions 3.1, fix a proximity parameter $\delta > 0$, it holds that*

$$\left\| \mathbb{E}_{\substack{u \sim Unif(\mathbb{S}^{d-1}), \\ Z \sim \Pi_{\boldsymbol{\theta} + \delta u}}}[g_\delta(\boldsymbol{\theta}; u, Z)] - \nabla \mathcal{L}(\boldsymbol{\theta}) \right\| = \|\nabla \mathcal{L}_\delta(\boldsymbol{\theta}) - \nabla \mathcal{L}(\boldsymbol{\theta})\| \leq \delta L$$

538 *Proof.* By Lemma E.1, we have

$$\mathbb{E}_{\substack{u \sim \mathrm{Unif}(\mathbb{S}^{d-1}), \\ Z \sim \Pi_{\boldsymbol{\theta} + \delta u}}}[g(\boldsymbol{\theta}; u, Z)] = \nabla \mathcal{L}_\delta(\boldsymbol{\theta})$$

539 Note that when $\mathcal{L}(\boldsymbol{\theta})$ is differentiable, we have

$$\nabla \mathcal{L}_\delta(\boldsymbol{\theta}) = \nabla \left[ \mathbb{E}_{w \sim \mathrm{Unif}(\mathbb{B}^d)} \mathcal{L}(\boldsymbol{\theta} + \delta w) \right] = \mathbb{E}_{w \sim \mathrm{Unif}(\mathbb{B}^d)} \nabla \mathcal{L}(\boldsymbol{\theta} + \delta w)$$

540 Then under Assumption 3.1, by linearity of expectation and Jensen's inequality, it holds that

$$\|\nabla \mathcal{L}_\delta(\boldsymbol{\theta}) - \nabla \mathcal{L}(\boldsymbol{\theta})\| = \left\| \mathbb{E}_{w \sim \mathrm{Unif}(\mathbb{B}^d)}[\nabla \mathcal{L}(\boldsymbol{\theta} + \delta w) - \nabla \mathcal{L}(\boldsymbol{\theta})] \right\| \leq \delta L.$$

541 $\qquad\square$

542 Note that if the performative risk only satisfies Lipschitz continuity, it is possible to apply similar
543 analysis to obtain convergence result for our algorithm. Informally, as $T \to \infty$, $\|\nabla \mathcal{L}_{\delta_k}(\boldsymbol{\theta})\|^2 \to 0$
544 at a rate of $\mathcal{O}((1+T)^{-\min\{1-\alpha, \alpha-3\beta\}})$, and $\mathcal{L}_{\delta_k}(\boldsymbol{\theta}) \to \mathcal{L}(\boldsymbol{\theta})$ at a rate of $\mathcal{O}((1+T)^{-\beta})$, where we
545 assume $0 \le 3\beta < \alpha < 1$. To find an $\epsilon$-stationary point of $\mu$-approximate performative risk function,
546 $\mathcal{O}(1/\epsilon^2 \mu^6)$ samples suffices.

547 **Corollary E.1.** *Under Assumption 3.1 and 3.2, for all $\boldsymbol{\theta} \in \mathbb{R}^d$, it holds that*

$$\|\nabla \mathcal{L}(\boldsymbol{\theta})\| \le 2\sqrt{LG}$$

548 *Proof.* Omitted. $\qquad\square$

549 **Lemma E.3. (Lipschitz Continuity of Decoupled Risk)** *Under Assumption 3.1, 3.2 and 3.3, it*
550 *holds that*

$$\left| \mathbb{E}_{Z \sim \Pi_{\boldsymbol{\theta}_2}} \left[ \ell(\boldsymbol{\theta}_1; Z) - \ell(\boldsymbol{\theta}_2; Z) \right] \right| \le 2(G L_2 + \sqrt{LG}) \|\boldsymbol{\theta}_1 - \boldsymbol{\theta}_2\| + \frac{L}{2} \|\boldsymbol{\theta}_1 - \boldsymbol{\theta}_2\|^2$$

551 *Proof.* Let $\mathcal{L}(\boldsymbol{\theta}_1, \boldsymbol{\theta}_2) := \mathbb{E}_{Z \sim \Pi_{\boldsymbol{\theta}_2}} \ell(\boldsymbol{\theta}_1; Z)$, then we have

$$
\begin{aligned}
\text{LHS} &= |\mathcal{L}(\boldsymbol{\theta}_1, \boldsymbol{\theta}_2) - \mathcal{L}(\boldsymbol{\theta}_2, \boldsymbol{\theta}_2)| \\
&\le |\mathcal{L}(\boldsymbol{\theta}_1) - \mathcal{L}(\boldsymbol{\theta}_2)| + |\mathcal{L}(\boldsymbol{\theta}_1, \boldsymbol{\theta}_2) - \mathcal{L}(\boldsymbol{\theta}_1, \boldsymbol{\theta}_1)| \\
&\le |\mathcal{L}(\boldsymbol{\theta}_1) - \mathcal{L}(\boldsymbol{\theta}_2) - \langle \nabla \mathcal{L}(\boldsymbol{\theta}_2) \,|\, \boldsymbol{\theta}_1 - \boldsymbol{\theta}_2 \rangle| + |\langle \nabla \mathcal{L}(\boldsymbol{\theta}_2) \,|\, \boldsymbol{\theta}_1 - \boldsymbol{\theta}_2 \rangle| + |\mathcal{L}(\boldsymbol{\theta}_1, \boldsymbol{\theta}_1) - \mathcal{L}(\boldsymbol{\theta}_1, \boldsymbol{\theta}_2)| \\
&\overset{(a)}{\le} \frac{L}{2} \|\boldsymbol{\theta}_1 - \boldsymbol{\theta}_2\|^2 + |\langle \nabla \mathcal{L}(\boldsymbol{\theta}_2) \,|\, \boldsymbol{\theta}_1 - \boldsymbol{\theta}_2 \rangle| + |\mathcal{L}(\boldsymbol{\theta}_1, \boldsymbol{\theta}_1) - \mathcal{L}(\boldsymbol{\theta}_1, \boldsymbol{\theta}_2)| \\
&\overset{(b)}{\le} \frac{L}{2} \|\boldsymbol{\theta}_1 - \boldsymbol{\theta}_2\|^2 + 2\sqrt{LG} \|\boldsymbol{\theta}_1 - \boldsymbol{\theta}_2\| + |\mathcal{L}(\boldsymbol{\theta}_1, \boldsymbol{\theta}_1) - \mathcal{L}(\boldsymbol{\theta}_1, \boldsymbol{\theta}_2)| \\
&= \frac{L}{2} \|\boldsymbol{\theta}_1 - \boldsymbol{\theta}_2\|^2 + 2\sqrt{LG} \|\boldsymbol{\theta}_1 - \boldsymbol{\theta}_2\| + \left| \int \ell(\boldsymbol{\theta}_1; z) \left( \Pi_{\boldsymbol{\theta}_1}(z) - \Pi_{\boldsymbol{\theta}_2}(z) \right) dz \right| \\
&\overset{(c)}{\le} \frac{L}{2} \|\boldsymbol{\theta}_1 - \boldsymbol{\theta}_2\|^2 + 2\sqrt{LG} \|\boldsymbol{\theta}_1 - \boldsymbol{\theta}_2\| + 2G \delta_{\text{TV}} \left( \Pi_{\boldsymbol{\theta}_1}, \Pi_{\boldsymbol{\theta}_2} \right) \\
&\le \frac{L}{2} \|\boldsymbol{\theta}_1 - \boldsymbol{\theta}_2\|^2 + 2\sqrt{LG} \|\boldsymbol{\theta}_1 - \boldsymbol{\theta}_2\| + 2G L_1 \|\boldsymbol{\theta}_1 - \boldsymbol{\theta}_2\| \\
&= 2 \left( \sqrt{LG} + G L_1 \right) \|\boldsymbol{\theta}_1 - \boldsymbol{\theta}_2\| + \frac{L}{2} \|\boldsymbol{\theta}_1 - \boldsymbol{\theta}_2\|^2
\end{aligned}
$$

552 where we use Assumption 3.1 in inequality (a), Corollary E.1 in inequality (b), Assumption 3.2 in
553 inequality (c), and Assumption 3.3 in the last inequality. $\qquad\square$

554 **Lemma E.4.** *Under Assumption 3.5, it holds that for all $0 \le \ell \le m$, $m \ge 1$*

$$\delta_{TV} \left( \mathbb{P}(Z_k^{(\ell+1)} \in \cdot | Z_k^{(0)}), \mathbb{P}(\tilde{Z}_k^{(\ell+1)} \in \cdot | Z_k^{(0)}) \right) \le L_2 \left\| \check{\boldsymbol{\theta}}_k^{(\ell)} - \check{\boldsymbol{\theta}}_k \right\| + \delta_{TV} \left( \mathbb{P}(Z_k^{(\ell)} \in \cdot | Z_k^{(0)}), \mathbb{P}(\tilde{Z}_k^{(\ell)} \in \cdot | Z_k^{(0)}) \right)$$

555 *Unfold above recursion leads to the following inequality,*

$$\delta_{TV} \left( \mathbb{P}(Z_k^{(m)} \in \cdot | Z_k^{(0)}), \mathbb{P}(\tilde{Z}_k^{(m)} \in \cdot | Z_k^{(0)}) \right) \le L_2 \sum_{\ell=1}^{m-1} \left\| \check{\boldsymbol{\theta}}_k^{(\ell)} - \check{\boldsymbol{\theta}}_k \right\|, \quad \forall m \ge 1.$$

556 *Proof.* Recall the notation $\check{\boldsymbol{\theta}}_k^{(\ell)} = \check{\boldsymbol{\theta}}_k^{(\ell)} + \delta_k u_k$, $\check{\boldsymbol{\theta}}_k = \check{\boldsymbol{\theta}}_k + \delta_k u_k$, and the fact that $Z_k = Z_k^{(0)} = \tilde{Z}_k^{(0)}$,
557 we have

$$
\begin{aligned}
2 \cdot \text{LHS} &= \int_{\mathsf{Z}} \left| \mathbb{P}(Z_k^{(\ell+1)} = z | Z_k^{(0)}) - \mathbb{P}(\tilde{Z}_k^{(\ell+1)} = z | Z_k^{(0)}) \right| dz \\
&= \int_{\mathsf{Z}} \left| \int_{\mathsf{Z}} \mathbb{P}(Z_k^{(\ell)} = z', Z_k^{(\ell+1)} = z | Z_k^{(0)}) - \mathbb{P}(\tilde{Z}_k^{(\ell)} = z', \tilde{Z}_k^{(\ell+1)} = z | Z_k^{(0)}) dz' \right| dz \\
&\le \int_{\mathsf{Z}} \int_{\mathsf{Z}} \left| \mathbb{T}_{\check{\boldsymbol{\theta}}_k^{(\ell)}}(z', z) \mathbb{P}(Z_k^{(\ell)} = z' | Z_k^{(0)}) - \mathbb{T}_{\check{\boldsymbol{\theta}}_k}(z', z) \mathbb{P}(\tilde{Z}_k^{(\ell)} = z' | Z_k^{(0)}) \right| dz' dz
\end{aligned}
$$

$$\leq \int_{\mathsf{Z}} \int_{\mathsf{Z}} \left| \mathbb{T}_{\check{\boldsymbol{\theta}}_k^{(\ell\ell}}(z', z) \mathbb{P}(Z_k^{(\ell)} = z' | Z_k^{(0)}) - \mathbb{T}_{\check{\boldsymbol{\theta}}_k}(z', z) \mathbb{P}(Z_k^{(\ell)} = z' | Z_k^{(0)}) \right| dz' dz$$

$$+ \int_{\mathsf{Z}} \int_{\mathsf{Z}} \left| \mathbb{T}_{\check{\boldsymbol{\theta}}_k}(z', z) \mathbb{P}(Z_k^{(\ell)} = z' | Z_k^{(0)}) - \mathbb{T}_{\check{\boldsymbol{\theta}}_k}(z', z) \mathbb{P}(\tilde{Z}_k^{(\ell)} = z' | Z_k^{(0)}) \right| dz' dz$$

$$\stackrel{(a)}{=} \int_{\mathsf{Z}} \mathbb{P}(Z_k^{\ell} = z' | Z_k^{(0)}) \int_{\mathsf{Z}} \left| \mathbb{T}_{\check{\boldsymbol{\theta}}_k}(z', z) - \mathbb{T}_{\check{\boldsymbol{\theta}}_k^{(\ell)}}(z', z) \right| dz dz'$$

$$+ \int_{\mathsf{Z}} \left[ \int_{\mathsf{Z}} \mathbb{T}_{\check{\boldsymbol{\theta}}_k}(z', z) dz \right] \left| \mathbb{P}(Z_k^{(\ell)} = z' | Z_k^{(0)}) - \mathbb{P}(\tilde{Z}_k^{(\ell)} = z' | Z_k^{(0)}) \right| dz'$$

$$\leq \int_{\mathsf{Z}} \mathbb{P}(Z_k = z' | Z_k^{(0)}) \cdot 2\boldsymbol{\delta}_{\mathrm{TV}} \left( \mathbb{T}_{\check{\boldsymbol{\theta}}_k}(z', \cdot), \mathbb{T}_{\check{\boldsymbol{\theta}}_k}(z', \cdot) \right) dz' + 2\boldsymbol{\delta}_{\mathrm{TV}} \left( \mathbb{P}(Z_k^{(\ell)} \in \cdot | Z_k^{(0)}), \mathbb{P}(\tilde{Z}_k^{(\ell)} \in \cdot | Z_k^{(0)}) \right)$$

$$\leq 2 \int_{\mathsf{Z}} \mathbb{P}(Z_k^{(\ell)} = z' | Z_k^{(0)}) dz' \cdot L_2 \left\| \check{\boldsymbol{\theta}}_k^{(\ell)} - \check{\boldsymbol{\theta}}_k \right\| + 2\boldsymbol{\delta}_{\mathrm{TV}} \left( \mathbb{P}(Z_k^{(\ell)} \in \cdot | Z_k^{(0)}), \mathbb{P}(\tilde{Z}_k^{(\ell)} \in \cdot | Z_k^{(0)}) \right)$$

$$= 2 \left[ L_2 \left\| \check{\boldsymbol{\theta}}_k^{(\ell)} - \check{\boldsymbol{\theta}}_k \right\| + \boldsymbol{\delta}_{\mathrm{TV}} \left( \mathbb{P}(Z_k^{(\ell)} \in \cdot | Z_k^{(0)}), \mathbb{P}(\tilde{Z}_k^{(\ell)} \in \cdot | Z_k^{(0)}) \right) \right] = 2 \cdot \mathrm{RHS}$$

where inequality (a) holds due to the (absolutely) integrable condition (which automatically holds for probability density functions and kernels), and Assumption 3.5 is used in the last inequality. $\square$