# OpenReview forum: "Two-timescale Derivative Free Optimization for Performative Prediction with Markovian Data"
_NeurIPS.cc/2023/Conference — Submitted to NeurIPS 2023_

### Official Review · Reviewer_3uod · 2023-06-11

**Soundness:** 3 good
**Presentation:** 3 good
**Contribution:** 3 good
**Rating:** 5
**Confidence:** 2

**Summary:**

The performative prediction problem targets real-world application using deployed models, and this method  studies the optimiazation method for such a scenario. Since directly optimiazing the non-convex objective is challenging, this paper utilizes weighted Markovian samples of states to estimate the gradient and update with designed timescale step size.

**Strengths:**

1. The paper is in general well-written.
2. The motivation is clear for tackling the targeted problem.
3. The theoretical derivation and understanding are well-presented.
4. The authors conduct various experiments settings to examine the the proposed method.

**Weaknesses:**

1. The experiments are conducted on toy examples and illustrate the efficacy of the method. The question would arise for applying the method for more challenging task setting. Also, how does the method perform in comparison with other strategies that also fit with such tasks?

2. What the limitation of the proposed method? As it targets online deployed models and involves the Markov Chain sampling, what is the computational cost?

**Questions:**

In general, the paper is well-written and would require background knowledge to understand the theoretical parts. My questions are listed:
1. L90:  "Let S be a (measurable) sample space, and µ, ν are two probability measures defined on S."
    and L114: "sample a vector µ uniformly from S^d-1". Does that imply that µ is uniformly distributed
    among S?

2. Given that Πθ(·) is unknown (L105-106) while the Markov kernel Tθ(·, ·) is based on Πθ(·), how would the Markov kernel conduct toward sampling?

3. L138-140: "Under this setting, suppose that the previous state/sample is Z, the next sample follows the distribution Z′ ∼ Tθ(Z, ·) which is not necessarily the same as Πθ(·)." It might needs some theoretical background to understand this, so my question is that since Πθ(·) serves a stational distribution given θ, why Z not converge to certain point and then Z′ should be sampled from the same distribution?





**Limitations:**

Please see the weaknesses and questions part.

---

> ### Author Rebuttal · Authors · 2023-08-09
>
> Thank you for the review. We address your concerns as follows.
>
> > The experiments are conducted on toy examples ... challenging task setting. Also, how does the method perform in comparison with other strategies that also fit with such tasks?
>
> We acknowledge that the experiments are mostly on toy examples due to the limited dataset available, especially for the challenging task of finding the performative optimal solution. This is a limitation found also in other existing works on performative prediction, e.g., (Perdomo et al., 2020), (Izzo et al, 2021), etc.
>
> We compared our approach to most of the available scheme under the setting with *Markovian data* with decision-dependent stationary distribution. Note that in Fig. 1, the benchmark scheme "SGD" is implementations which ignore such decision-dependent distribution structure, "Pure DFO" ignores the Markov data setting, "DFO ($\lambda=0$)" is *similar* to the algorithm suggested in (Ray et al., 2022).
>
> > What is the limitation of the proposed method? What is its computational cost?
>
> One of the limitations is the algo's dependence on dimension $O(d^2)$ in the sample complexity. Note that similar drawback is shared by other DFO algorithms. However, we believe that the use of DFO in the performative prediction setting has outweighed this drawback since the algorithm needs not to form an estimate about the distribution as in prior works such as (Izzo et al., 2021).
> Since the Markov chain sampling would not incur extra calculation, the computational cost is in the same order as the number of samples (off by at most a dimension $d$).
>
> > L90: "Let S be a (measurable) sample space, and $\mu$, $\nu$ are two probability measures defined on ${\cal S}$." and line 114: "sample a vector $\mu$ uniformly from $\mathbb{S}^{d-1}$". Does that imply that $\mu$ is uniformly distributed among $\mathcal{S}$?
>
> Note that $\mu$ in L90 is the probability measure on ${\cal S}$, which is a sample space; while $u$ in L114 is a vector sampled from $\mathbb{S}^{d-1}$, which is the boundary of a $d$-dimensional unit ball as defined in L93. Note that ${\cal S}, \mathbb{S}^{d-1}$ are different objects.
>
> The latter sentence in L114 states that the vector $u$ is drawn uniformly from $\mathbb{S}^{d-1}$.
>
> > Given that $\Pi_{\theta}(·)$ is unknown (L105-106) while the Markov kernel $T_{\theta}(·, ·)$ is based on $\Pi_{\theta}(·)$, how would the Markov kernel conduct toward sampling?
>
> In the performative prediction setting considered by this paper, the Markov kernel is *not known* to the algorithm, but is rather treated as an oracle that accepts queries and outputs samples to be used in the algorithm. One can imagine that the Markov kernel models the adaptation process of the population who is interacting with the learner that executes the algorithm. In this case, the learner does not possess knowledge of the Markov kernel, but merely the samples taken from it.
>
> > L138-140: "Under this setting, suppose that the previous state/sample is Z, the next sample follows the distribution $Z^\prime∼ T_{\theta}(Z, ·)$ which is not necessarily the same as $\Pi_{\theta}(·)$." It might needs some theoretical background to understand this, so my question is that since $\Pi_{\theta}(·)$ serves a stational distribution given θ, why Z not converge to certain point and then Z′ should be sampled from the same distribution?
>
> In general, the samples drawn from a Markov chain do not follow the stationary distribution, except for degenerate cases such as those where $T_{\theta}(z, ·)$ is identical for all $z$, or when the sample $z$ is already from a stationary distribution. It is only after a certain time (i.e., mixing time) then the process would generate samples that are close to the stationary distribution (e.g., see Assumption 3.4). By `Z converge to certain point, and then Z′ should be sampled from the same distribution`, we believe that the reviewer is referring to the situation when the Markov chain has converged to the stationary distribution. In our setting, when $\theta$ is fixed, Assumption 3.4 guarantees that the Markov chain has a unique stationary distribution (note it only converges to a *distribution*, but not a *deterministic point*). As this $\theta$ is even *time varying* in our performative prediction setting as the decision model is updated and deployed at **each** epoch/iteration, this further adds to the complexity of our analysis.

---

> > ### Comment · Reviewer_3uod · 2023-08-17
> >
> > I thank you for the feedback provided by the authors. After reading the responses, I tend to keep my rate score.

---

### Official Review · Reviewer_bs2h · 2023-07-05

**Soundness:** 4 excellent
**Presentation:** 2 fair
**Contribution:** 1 poor
**Rating:** 5
**Confidence:** 4

**Summary:**

This paper proposes a derivative-free two-time scale  for performative prediction problem. The two-timescale facilitates a faster accumulation of samples to compute a gradient with smaller bias. For smooth nonconvex objective, the work proves a iteration complexity of $O(\epsilon^{-3})$ for the $\epsilon$-staitonary point. They illustrate results via numerical experiments.

**Strengths:**

1. This paper deals with an important problem: derivative-free optimization of nonconvex function under performative prediction setup.

2. The convergence rate seems reasonable.

3. The experiments are well-motivated.

**Weaknesses:**

1. **The applications are shown for squared loss. But it is not a bounded loss and does not satisfy Assumption 3.2. This leads me to believe that Assumption 3.2 is made for the convenience of theoretical analysis.**

In fact, **one of the major challenges in the decision-dependent noise or state-dependent Markov noise setup is that the algorithm is often not stable** for unconstrained optimization (see [1,2] below). Assumption 3.2 helps to avoid this issue.

**So Assumption 3.2 is crucial for Lemma E.3 and Lemma B.i (i=1,2,3,4). Then the theoretical analysis of the algorithm does not apply to squared loss which is one of the most popular loss functions.**

2. The proof techniques required are pretty similar to Wu et al., 2020 ([3] below). In fact, **the main result is almost same as Corollary 4.9 of Wu et al., 2020. It's just that Corollary 4.9 of Wu et al., 2020 is wrapped in the cover of reinforcement learning but the algorithm, underlying proof techniques, and result are same.**  Wu et al., 2020 does achieve a faster rate of $\tilde{O}(\epsilon^{-2.5})$ although I guess here the poor rate is due to derivative-free algorithm.


[1] Liang, Faming. "Trajectory averaging for stochastic approximation MCMC algorithms." (2010): 2823-2856.

[2] Andrieu, Christophe, Éric Moulines, and Pierre Priouret. "Stability of stochastic approximation under verifiable conditions." SIAM Journal on control and optimization 44, no. 1 (2005): 283-312.

[3] Wu, Yue Frank, Weitong Zhang, Pan Xu, and Quanquan Gu. "A finite-time analysis of two time-scale actor-critic methods." Advances in Neural Information Processing Systems 33 (2020): 17617-17628.

**Questions:**

1. It is not clear to me whether the derivative-free approach helps to deal with state-dependence of the Markov chain like mentioned in line 105-112? In other words, say there is a gradient oracle such that $g_{k}^{(m)}=\nabla l(\theta_k^{(m)},Z_{k}^{(m)})$. What will happen to the convergence rate of the algorithm then?

2. Please highlight the technical novelties required compared to Wu et al., 2020 (apart from the zeroth-order approximation). See point 2 of Weakness for details.

I may change my score depending on the reply.

---

> ### Author Rebuttal · Authors · 2023-08-09
>
> Thank you for the comments which inspired us to explore our contributions more. Referring to the _General Responses_, we reiterate that the our DFO algo is new which has not been analyzed before, nor is it studied with Markov data, further it is motivated by **practical** performative prediction (e.g., it does not require knowledge of distribution, unlike in RL where effect of policy on state transition is partially known). Below, we wish to clear the confusions about our contributions by responding to the weaknesses/questions.
>
> > The applications are shown for squared loss ... does not satisfy Assumption 3.2. And the theretical analysis ...
>
> As the reviewer has rightfully pointed out, A3.2 is violated by squared loss in our experiments. As a common design, its use in related prior works such as (Izzo et al., 2021) is the main reason why we included such example. That said, we argue that our analysis can still yield meaningful insights.
>
> First, we observe that in practice both $\theta, z$ are bounded throughout the algo. As A3.2 is only used in bounding the terms generated from the algo., our results naturally follow. This is evident from Fig. 1 where the DFO has convergence behavior accurately predicted by the analysis.
>
> Second, it is possible to _approx._ an unbounded loss function by a bounded one. An example is inverted Gaussian trick: e.g., $\ell(\theta,z) = (\theta+z)^2/2$ can be approx. as $\tilde{\ell}(\theta,z) = 1-\exp(-(\theta+z)^2/(2\sigma^2))$ for $\sigma>0$ controlling the approx quality - note it satisfies A3.1, A3.2. As a complimentary example, in the attached PDF, we provide a simple experiment with the above loss, $\Pi_{\theta} = {\cal N}(\theta,1)$ and samples drawn from AR. Similar behaviour as Fig. 1 is observed - DFO converges at $O(1/T^{0.41})$ as predicted by analysis. The above example will be included in the final version.
>
> > Similarity of proof to Wu et al., 2020
>
> Our proposed algo embodies a DFO update for handling the lack of knowledge about distribution shift as well as a sample accumulation scheme to reduce bias. Wu et al. (2020) studied an actor-critic (AC) algo that has the policy gradient for actor-update and a coupled TD learning step for critic-update. While both algo use 2-timescale (2TS) steps, the latter's uses are completely different - our DFO algo. uses it to control variance in gradient estimation, while AC algo. uses it to produce an accurate policy evaluation. Our analysis is different due to the different structure of the problem and algorithm.
>
> The main (& only) similarity between our analysis & Wu et al. comes from the idea of constructing a fake Markov chain (MC) in our Lemma B.2 as A.2.2, Step 2 of Wu et al. We borrowed this technique to separate biases due to the drifting $\theta$ and that due to the mixing of MC. We explain below at pt.2 that even this technique is used under different context. In addition to the DFO strategy (which already led to a very different analysis), our analysis differs from Wu et al. in at least 4 aspects:
>
> - Our analysis, which begins from Lemma 4.1, is based on analysis of **smooth & non-convex** optimization of ${\cal L}(\theta)$, in which the quantities $I_1(t),...,I_4(t)$, and focus on $||\nabla {\cal L}(\theta)||$. Herein, we _do not exploit any structure_ such as convexity of loss function. For Wu et al., their analysis is split into two parts - the first part is on the smooth, non-concave avg. reward function is proven under (4.2) on the critic error, the second part controls the critic error done in A.2.2. This critic "subproblem" is **strongly convex** for fixed policy, as seen in their A.2.2.
> - The fake MC mentioned is used for controlling $I_2(t)$ in our paper, and $I_2$ in Wu et al. The terms analyzed are different - our $I_2(t)$ sums up inner product of $\nabla L(\theta)$ and the error of accumulated _gradient estimator_ and its copy based on a stationary distribution induced by $\check{\theta}_k$; while $I_2$ in Wu et al. sums up inner product between critic error $w_t - w_t^*$ & TD update error w.r.t. mean field. Without convexity, similar term to $w_t^*$ cannot be found in our setting. This results in a different form of upper bound in our (21) and (A.6) in Wu et al.
> - The role of 2TS-step are different. In our analysis, the coupling btw $\eta_k, \delta_k$ appear in all $I_2(t),...,I_4(t)$. For Wu et al., it directly control the drift in the critic as reflected in their $I_3$.
> - Due to the different algo structure, our terms $I_3(t),I_4(t)$ entail different forms than those analyzed as $I_3, I_4$ in Wu et el.
>
> We remark that the technique of splitting terms in our Lemma 4.1 and in (A.4) of Wu et al. is common in SA analysis, e.g., (Karimi et al., 2019), (Li and Wai, 2022), (Roy et al., 2022).
>
> > Does the derivative-free approach help to deal with state-dependence of the Markov chain? If there is a gradient oracle such that $\nabla \ell(\theta_{k}^{(m)}, Z_{k}^{(m)})$. What will happen to the convergence rate of the algorithm then?
>
> We emphasize DFO is motivated by the **lack** of knowledge on the state-dependent distribution in performative prediction (e.g., privacy concern of population), which is required otherwise for **gradient-based** algorithms. It was _not_ designed to deal with the state-dependence of MC. The latter is rather an issue to be handled, and we proposed a sample-accumulation scheme for handling it.
>
> When the said gradient oracle is available, (Li and Wai, 2022) showed that it can be used for SA which finds the *performative stable* solution if $\ell(.,z)$ is strong convex (+ other assumptions). For finding a _performative optimal_ solution, it is not clear if such knowledge is useful since the true gradient derived in (3) requires a correction term. As such, having such an oracle may not improve the current rate unless additional assumptions on distribution map are added as done in (Izzo et la., 2021).

---

> > ### Comment · Reviewer_bs2h · 2023-08-19
> > **Thanks for the reponse.**
> >
> > I am happy with the responses. I have increased my score.

---

### Official Review · Reviewer_vEJR · 2023-07-05

**Soundness:** 4 excellent
**Presentation:** 4 excellent
**Contribution:** 3 good
**Rating:** 7
**Confidence:** 4

**Summary:**

The paper studies performative prediction when the data is stateful, in particular generated via a controlled Markov chain, and the learner's loss is possibly nonconvex. The paper develops a two-timescale derivative-free optimization algorithm for this setting and shows a O(1/eps^3) sample complexity for finding a point with squared gradient norm at most eps.

**Strengths:**

There aren't many convergence results in performative prediction for nonconvex settings, so this is one strength. The stateful setting has been studied before in several papers, but this paper studies this setting under quite a bit of generality. The treatment of the stateful setting in this paper is probably my favorite treatment of the setting in the literature. The paper is very clearly written and easy to follow, also providing intuition for the ideas behind the analysis, which I appreciated.

**Weaknesses:**

There are some limitations to the conceptual novelty, in the sense that a similar algorithm has been studied outside of performative prediction, and the controlled Markov chain model for the distribution map has been studied. It would be good to be more precise about the differences to the recent works in performative prediction studying the stateful setting (bottom of page 2).

The problem is perhaps arguably a bit niche since it goes away if the performativity is not stateful, which is the most commonly studied observation model in performative prediction. In that case, a simple application of the Flaxman et al. "gradient descent without a gradient" algorithm suffices. The issue in this paper is that we can't sample from the stationary distribution directly so a naive application of the Flaxman et al. algorithm doesn't work. All this being said, the stateful setting is very well-motivated and deserves its own analyses, and this paper gives a clever solution.

**Questions:**

1. Can you please clarify the differences to prior work studying stateful performative prediction?
2. Do you have a sense of why SGD has these strange dips in Figure 1?
3. In lines 181-182, you say that the geometric dynamically environment of Ray et al. (2022) constitutes a special case of your assumptions. Can you give more natural examples of stateful distribution maps that satisfy your assumptions but are different from Ray et al.?

---

> ### Author Rebuttal · Authors · 2023-08-09
>
> We are glad that the reviewer liked our paper recongnizing the theoretical contributions of performative prediction problem under the nonconvex, stateful settings. Our point-to-point responses to concerns on weakness and questions are:
>
> > Discussion about the differences between recent performative prediction literature and this paper.
>
> We agree and will extend the discussions on prior works about performative prediction in the final version (if space allows).
>
> > In simulation part, why does the SGD have these strange dips in Figure1?
>
> The SGD applied in Fig. 1 is supposed to converge to a **performative stable** solution that makes the first term in (3) zero, but not the second term therein, i.e., it is not a stationary point to the performative risk ${\cal L}(\theta)$. While the exact reason for the dip in the beginning of iterations is unknown, we suspect it is due to that SGD making the first term in (3) zero in the first phase, while in the latter phase, the second term in (3) increases.
>
> > In lines 181-182, you say that the geometric dynamically environment of Ray et al. (2022) constitutes a special case of your assumptions. Can you give more natural examples?
>
> One example that is included by our assumptions but not by Ray et al., 2022, is that of the Markov Decision Process (MDP) which is commonly found in RL. Here, the decision $\theta$ controls the conditional probability of choosing the next action under the current state. Especially when the state space $Z$ is finite, the state transition of the MDP cannot be described using the geometric decay model in Ray et al. Note that this is relevant to the performative prediction setting as $\theta$ can be regarded as the action policy followed by the population.
>
> Another example is an AR(2) map modified from (Ray et al, 2022) as ${\cal T}(p,p^-,\theta) = \lambda( p+p^-)/2 + (1-\lambda) \Pi_{\theta}$, where $p, p^-$ refer to the current and previous distributions, respectively. The evolution described no longer fits into the geometric decaying envir in (Ray et al, 2022); yet it can be described using a Markov chain satisfying our A3.4 with concatenated states.

---

> > ### Comment · Reviewer_vEJR · 2023-08-11
> > **Thank you for the response**
> >
> > Thank you for the response. The examples at the end are interesting and helpful.

---

### Official Review · Reviewer_pcvM · 2023-07-06

**Soundness:** 4 excellent
**Presentation:** 3 good
**Contribution:** 2 fair
**Rating:** 5
**Confidence:** 3

**Summary:**

This paper is on performative prediction i.e. a setting where the data distribution changes in response to the predictions of a learned model. The prototypical example of such a setting is where a learned model is used to make loan decisions, and then people or companies adjust their behavior based on knowledge of the learned model in order to secure more favorable loan terms. This paper extends prior work on performative prediction to the setting where the data distribution follows a controlled Markov process, where the control is given by the predictive model. While this setting has been studied before, this paper removes previous structural assumptions on (1) the loss function used to evaluate the model and (2) the dependence of the data distribution on model. Notably the loss function is not assumed to be convex, and so convergence is established to a stationary point (i.e. a point where the magnitude of the gradient is small).
The algorithm designed is based on standard methods for derivative-free optimization, with a two-timescale step-size modification that updates model parameters more slowly than the generation of gradient estimates in order to deal with high variance in the gradient estimator.

**Strengths:**

The extension of algorithms for performative prediction to the setting without convexity of the loss expands the range of techniques for such problems. The paper clearly discusses the differences with prior work and gives a concise and intuitive overview of the modifications required to make derivative-free optimization work for this setting.

**Weaknesses:**

While I do appreciate the generality of extending algorithms for performative prediction beyond convex losses (as well as beyond the mixture dominance assumption on the data distribution), there are a couple of limitations that arise from assuming so little about the loss and data distribution.

1. The algorithm converges to a stationary point of the performative risk, rather than to a point that approximately minimizes the preformative risk. Of course this is necessary in the completely unstructured setting considered here, but it is clearly a weaker statement than e.g. the initial work of Perdomo et. al. that showed (for strongly convex losses) convergence of repeated risk minimization to a point that is close to an actual minimizer of the performative risk. This paper should state more clearly that such a strong result cannot be hoped for in the setting considered. In particular, the claim on line 43 seems somewhat misleading as written.

2. Removing the convexity assumption seems to come at the cost of introducing an additional assumption. In particular, Assumption 3.3 requires that the distribution map $\Pi_{\theta}$ is Lipschitz with respect to the total variation distance, rather than the Wasserstein 1 distance used in prior work. This is a **much stronger** assumption in many natural settings e.g. the mean of $n$-independent random $\pm 1$ valued variables has a distribution that converges in Wasserstein 1 distance to the Gaussian distribution at a rate of $\frac{1}{\sqrt{n}}$, but has the maximum possible total variation distance of 1 from the standard Gaussian distribution.

3. The algorithm has a slower convergence rate than those in prior work, as the authors show is necessary in this highly general setting. The original paper introducing performative prediction shows that repeated risk minimization converges at a linear rate, and thus allows us to leverage whatever fast optimization method fits the particular problem for each risk minimization step. In contrast, the algorithm in this paper requires us to essentially sample many, many random perturbations of the model and then deploy/evaluate each one in order to obtain gradient estimates, making it highly impractical for real-world problems.

To summarize, the generality of the setting has several drawbacks in terms of the solution quality, performance of the algorithm, and most notably the additional assumption described in (2) above. While some of these drawbacks are provably necessary (again with the notable exception of the issue in (2)), taken together they suggest that perhaps this complete lack of structure is not a particularly good model of the types of problems for which performative prediction is interesting.


**Questions:**

1. Is the Lipschitz assumption with respect to the total variation distance necessary in this setting?
2. Are there any intermediate models with assumptions weaker than strong convexity of the loss, but stronger than the assumptions made in this paper that could also be interesting?

**Limitations:**

Yes.

---

> ### Author Rebuttal · Authors · 2023-08-09
>
> Thank you for the careful review. As mentioned in the general response, we emphasize that our contributions lie on proposing a practical algorithm for performative prediction (aiming at finding the performative optimal solution) with limited knowledge assumed on the learner's side. Below we shall address your concerns on weaknesses and questions raised.
>
> > This paper should state more clearly that such a strong result cannot be hoped for ... the claim on line 43 seems somewhat misleading as written.
>
> We agree that the sentence in L43 is somewhat misleading. It will be revised to "This paper seeks to approximate an performative optimal solution without ..."
>
> > Lipschitzness with respect to the TV distance vs Wasserstein-1 distance.
>
> Thanks for raising this point. Our answer is affirmative. Assumption 3.3 is only used in the proof of Lemma E.3 to bound the term in L549. At the cost of imposing an additional assumption on the partial Lipschitzness, i.e.,
>
> $$|\ell(\theta,z) - \ell(\theta,z')| \leq L_0 ||z-z'||, \forall \theta,z,z'$$
>
> we can relax A3.3 to be based on the Wasserstein-1 distance, i.e.,
>
> $$W_1(\Pi_{\theta_0}, \Pi_{\theta_1}) \leq L_1 || \theta_0 - \theta_1 ||, \forall \theta_0, \theta_1.$$
>
> To use the above setting in Lemma E.3, observe
>
> $$\begin{aligned}|{\cal L}(\theta_1,\theta_1) - {\cal L}(\theta_1,\theta_2)| = |\mathbb{E}_{Z\sim\Pi({\theta_1}), Z'\sim\Pi({\theta_2})}[\ell(\theta_1,Z) - \ell(\theta_1,Z')]| \end{aligned}
> $$
>
> $$
> \qquad \qquad \qquad \quad  \qquad \leq L_0 \mathbb{E}_{Z \sim \Pi(\theta_1), Z' \sim \Pi(\theta_2)}[||Z-Z'||] $$
>
> which can be further bounded by $L_0 L_1 ||\theta_1-\theta_2||$ using the Wasserstein-1 distance assumption. Substituting into L551 (after (b)) yields the final bound of $2(\sqrt{LG}+L_0 L_1)||\theta_1 - \theta_2|| + (L/2)||\theta_1-\theta_2||^2$.
>
> We apologize for not including the more complete result above. It will be included in the final version.
>
> > The algorithm has a slower convergence rate than those in prior work,... making it highly impractical for real-world problems
>
> We note that faster rate than $O(1/\epsilon^3)$ is only available for problems with richer structure - such as a rate of $O(1/\epsilon^2)$ for e.g., w/ **state dependent** distribution under the mixture dominance condition (which implies strongly convex ${\cal L}(.)$) - Ray et al., 2022, w/ **state independent** distribution - Ghadimi & Lan, 2012. As explained in Sec 4.2, we believe that improving the rate may not be possible.
>
> Regarding practicality, we note that using DFO as our backbone design can eliminate the need to estimate the gradient of the log-distribution $\nabla \log \Pi_\theta(Z)$ in (3). In performative prediction, learning the latter is almost the same as learning the responses from the population which is much more difficult unless further assumptions are made (e.g., Izzo et al., 2021). Together with the ability to work with Markovian samples for **stateful** response, we believe that our results are sufficiently useful and practical. Please refer to the _General Responses_.
>
> >  ...the initial work of Perdomo et. al. that showed (for strongly convex losses) convergence of repeated risk minimization to a point that is close to an actual minimizer of the performative risk.
>
> > The original paper introducing performative prediction shows that repeated risk minimization (RRM) converges at a linear rate ...
>
> The RRM analyzed by (Perdomo et al., 2020) requires minimizing the expected risk **exactly** which requires infinite samples in general. It also only finds the performative stable point which may not exist if the distribution is sensitive to the decision model changes (i.e., when $L_1$ is large), together with other assumptions such as strong convexity of $\ell(.)$. None of these conditions are required by our paper.
>
> > Intermediate models with assumptions weaker than strong convexity of the loss, but stronger than the assumptions made in this paper
>
> Thank you for the suggestion. We agree that intermediate models that are weaker than strong convexity as in the mixture dominance condition of (Miller et al., 2022) would be interesting and is a promising direction for future research. Indeed, we notice the following related work along this line
>
> Y. Zhao, Optimizing the Performative Risk under Weak Convexity Assumptions, arXiv, 2022.
>
> yet it does not suggest a practical algorithm. At last, we emphasize that the focus of our work is to study a practical scheme based on DFO that attempts at approximating the performative optimal solution without making strong assumptions on the distribution, objective function, nor the data sampling.

---

> > ### Comment · Reviewer_pcvM · 2023-08-16
> >
> > Thank you for clarifying that Lipschitzness with respect to the total variation distance is not necessary for the results to hold. I will increase my score to 5.

---

### Official Review · Reviewer_NH9K · 2023-07-07

**Soundness:** 3 good
**Presentation:** 3 good
**Contribution:** 3 good
**Rating:** 6
**Confidence:** 2

**Summary:**

The paper considers the performative prediction problem, where the goal of learner is to optimize the expectation of the known loss function over a decision-dependent unknown data distribution that evolves according to an underlying controlled Markov chain. Authors presents a stochastic derivative-free optimization algorithm $DFO(\lambda)$ that achieves $O(d^2/\varepsilon^3)$ sample complexity using gradient accumulation mechanism and two-timescale diminishing step-sizes.

**Strengths:**

- The presented algorithm achieves a $O(1/\varepsilon^3)$ sample complexity under a challenging setting of Markovian data.
- The other possible types of gradient estimator were considered that creates a more complete picture in the presented setup.
- The presentation of the paper is clear.

**Weaknesses:**

- No presented lower bound for this problem, thus it is not clear is the presented rates improbable or not.

**Questions:**

- What could be challenges to generalize the proposed algorithm to the milder assumption on the Markov chain, i.e. V-ergodicity instead of the uniform ergodicity.
- What is the bias of the first version of two-point estimate $g_{2pt-I}$? On which variables it depends in the worst case?
- Is there some stronger assumptions on the Markov chain under which it is possible to recover $O(1/\varepsilon^2)$ rates as in i.i.d. case?
- Can be this algorithm applied to the reinforcement leaning problem?

**Limitations:**

This is a theoretical paper that does not need to address the potential societal impact.

---

> ### Author Rebuttal · Authors · 2023-08-09
>
> Thank you for the review. We address your concerns as follows.
>
> > lower bound for the problem
>
> We are not aware of any such lower bound result for the **decision-dependent** stochastic optimization problem (1) without access to $\Pi_{\theta}$. In fact, to our best knowledge, even in the setting when the objective function is **decision-independent**, e.g., ${\cal L}(\theta) = E_{Z \sim \Pi}[ \ell(\theta;Z)]$, and i.i.d. samples of noisy function evaluation are available, the complexity lower bound for _non-convex_ and smooth optimization using derivative free stochastic algorithm is not known to our best knowledge. For _strongly convex_ and smooth optimization, a lower bound of $\Omega(1/\epsilon^2)$ measured w.r.t. optimality gap of objective value has been shown in (Jamieson et al., 2012).
>
> > challenges to generalize to milder assumption on the Markov chain.
>
> To adapt the current proof to milder settings on Markov chain such as V-uniform-ergodicity, a key technical challenges lies with the proof of Lemma 4.2 & B.2. At line 428, V-uniform-ergodicity will modify the upper bound to $G M \rho^m V(\tilde{Z}_k^0)$ that depends on the initial state at each epoch of the algorithm. Accumulating these terms may result in a nontractable sum. In any cases, the reviewer has raised a good point here, which we will consider in the future work.
>
> > What is the bias of the first version of two-point estimate $g_{2pt-I}$? On which variables it depends in the worst case?
>
> For simplicity, let us explain it using the stationary distribution setting in Sec. 4.1. As discussed in (4), the first term $\frac{d}{\delta}\ell(\theta + \delta u;Z)u$ of the estimator is almost unbiased. Now, the bias is due to the second term $\frac{d}{\delta}\ell(\theta;Z)u$ since $Z$ is drawn from a perturbated distribution $\Pi_{\theta+\delta u}$, resulting in the expected value $E_u [ E_{ Z \sim \Pi_{\theta+\delta u} } [ \ell(\theta; Z) ] ]$ which is **not** the gradient of ${\cal L}_\delta(\theta)$. In fact, this second term introduces an overall bias in the order of $O(1/\delta)$.
>
> > stronger assumptions on the Markov chain under to recover $O(1/\epsilon^2)$ rates as in i.i.d. case?
>
> We suspect that a stronger assumption on the MC would not help. To achieve an $O(1/\epsilon^2)$ rate using DFO, we are not aware of any approaches other than using the estimator $g_{2pt-I}$ due to Ghadimi et al., where the two-point evaluation helps to reduce variance in gradient estimation. However, as explained in Sec 4.1, such an estimator does not work for decision-dependent setting even with samples drawn iid from stationary distribution.
>
> > Can the proposed algorithm be applied to reinforcement learning?
>
> Our DFO algo design is motivated by the performative prediction setting where the decision dependent distribution $\Pi_{\theta}$ is **unknown** and can be arbitrary. On the other hand, in RL problems such as (stochastic) policy optimization, the parameter controlling the policy is usually well specified and **known**. In other words, the latter problem contains richer structure that can be leveraged for a more efficient algorithm. Nevertheless, we believe that techniques such as the sample accumulation in this paper for bias reduction can be insightful.

---

> > ### Comment · Reviewer_NH9K · 2023-08-18
> >
> > I would like to thank the authors for their answers and additional discussions. The comments have addressed my questions and I decide to keep my score.

---

### Author Rebuttal · Authors · 2023-08-09

We thank all the five reviewers for their careful reviews. We summarize our responses and highlight the main contributions of our paper.

The performative prediction (PP) problem constitutes a new class of stochastic optimization problems with decision dependent distribution. This is challenging as (i) problem (1) is non convex in general, (ii) the learner lacks info. about the (stochastic) gradient as $\Pi_{\theta}$ is not known, (iii) the population is stateful as adaptation can be slow. The latter two issues which result from practical concerns unique to PP have not been discussed in the literature.

We are thus motivated to propose a derivative free opt (DFO) scheme to (1). Thanks to the DFO design, the learner only needs noisy samples of the loss values that are naturally available without requiring knowledge of the distribution $\Pi_{\theta}$, making it a practical solution suitable for PP. This is unlike in RL (which may appear to be similar to PP) where the state-dependent distribution is partially known to the learner, e.g., policy opt typically needs $\nabla_{\theta} \log p_{\theta}(a|s)$. Besides, we make the following technical contributions:
- We show that direct application of DFO does not converge (see Fig. 1) under Markov data with a 1-point estimator design that works with iid data. To avoid burning in of samples as done in (Ray et al., 2022), we propose a sample accumulation scheme (L6-10 in Algo. 1) that utilizes every samples collected from the Markov chain but still produces a near bias-free gradient estimator. This scheme is found to be more sample efficient in practice.
- We show in Sec 4.2 that the $O(1/\epsilon^3)$ rate is likely tight by demonstrating that the 2-point estimators, typically used for getting $O(1/\epsilon^2)$ rate for DFO, are inapplicable,
- We develop finite-time analysis under (adaptive) Markov data. Our analysis has been inspired by (Wu et al., 2020) which introduces a "fake" Markov chain with the control parameter fixed at the epoch's initialization. However, as our algorithm embodies a DFO gradient estimator and a unique sample accumulation scheme, we had to develop a new set of analysis to control the respective error terms.
- Appendix D also provides an extended analysis for the case when ${\cal L}(.)$ is not smooth, i.e., without A3.1.

In response to the reviewers' comments, we strengthen some of our results as follows (to be included in the final version):
- If we further assume $\ell(\theta, z)$ to be Lipschitz w.r.t. $z$, then A3.3 on the Lipschitzness of distribution shift with TV distance can be relaxed to a weaker notion relying on the Wasserstein-1 distance, see the response to pcvM.
- We constructed two examples of stateful population that are covered by our setup modeled using controlled Markov chain, while not covered by the setup based on geometrically decay in (Ray et al., 2022), see the response to vEJR.
- We conducted additional experiment on an inverted Gaussian loss, i.e., a bounded function that satisfy all assumptions required in our analysis, see the attached PDF and response to bs2h.

Our work provides a practical solution to performative prediction and can shed further light in the design/analysis of DFO algorithms. We are happy to further engage with the reviewers to address any other concerns.

---

### Decision · Program_Chairs · 2023-09-21

**Decision:**

Reject

**Comment:**

While the work seems to contribute an interesting and potentially novel algorithm to derivative-free optimization, the reviewers are in agreement that the technical assumptions (especially 3.2 and 3.3) required for the results to hold are too restrictive, and that the key steps in the analysis are very similar to Corollary 4.9 of Wu et al., 2020. Therefore, the authors should expend more time to relax key technical assumptions required for their theory, or otherwise identify key points of technical departure in their analysis before it is publishable.